# TESTING FAIRNESS WITH UTILITY TRADE-OFFS: A WASSERSTEIN PROJECTION APPROACH

## ABSTRACT

Ensuring fairness in data-driven decision-making has become a central concern across domains such as marketing, lending, and healthcare, but fairness constraints often come at the cost of utility. We propose a statistical hypothesis testing framework that jointly evaluates approximate fairness and utility, relaxing strict fairness requirements while ensuring that overall utility remains above a specified threshold. Our framework builds on the strong demographic parity (SDP) criterion and incorporates a utility measure motivated by the potential outcomes framework. The test statistic is constructed via Wasserstein projections, enabling auditors to assess whether observed fairness–utility trade-offs are intrinsic to the algorithm or attributable to randomness in the data. We show that the test is computationally tractable, interpretable, broadly applicable across machine learning models, and extendable to more general settings. We apply our approach to multiple real-world datasets, offering new insights into the fairness–utility trade-off through the perspective of statistical hypothesis testing.

## 1 INTRODUCTION

Over the past decade, ensuring fairness in data-driven decision-making has become a critical concern across many domains, including personalized marketing, lending, and healthcare (Kallus & Zhou, 2021; Richards et al., 2016; Liu et al., 2019; Kumar et al., 2022; Ahmad et al., 2020; Chen et al., 2023; Giovanola & Tiribelli, 2023; Bertsimas et al., 2012; Manski et al., 2023; Qi, 2017). A substantial body of research has sought to formalize fairness through constraints on predictive models or algorithms (Gardner et al., 2019; Alikhademi et al., 2022; Pleiss et al., 2017; Jacobs & Wallach, 2021; Taskesen et al., 2021; Navarro et al., 2021), aimed at safeguarding individuals or groups from discriminatory treatment or policies (Chouldechova, 2017; Imai & Jiang, 2023; Kizilcec & Lee, 2022).

However, imposing fairness constraints often entails trade-offs with utility. For instance, Mehrotra et al. (2018) documents a tension between supplier fairness and consumer satisfaction in recommender systems for two-sided online platforms. Another example is the accuracy–fairness trade-off in image classification and representation learning, examined by Dehdashtian et al. (2024), who develop a method to numerically quantify this trade-off for specific prediction tasks and group fairness criteria, thereby introducing a new evaluation framework for computer vision representations. Several other studies have also shown that achieving absolute fairness while preserving utility is impossible in many applications, as fairness constraints inevitably reduce the performance of the targeted utility (Mitchell et al., 2021; Cooper et al., 2021, etc.). Moreover, many existing methodologies for evaluating fairness–utility trade-offs also tend to be overly task-specific (Sacharidis et al., 2019; Dehdashtian et al., 2024, etc.).

These observations motivate a more nuanced approach to algorithmic fairness with utility trade-off — one that seeks to ensure approximately fair outcomes across protected groups while explicitly preserving an adequate level of overall utility. Indeed, there has been growing interest in recent years in pursuing algorithmic fairness through frameworks that explicitly account for trade-offs with utility (Ge et al., 2022; Rodrigues & Casadevall, 2011; Plecko & Bareinboim, 2025; Chester et al., 2020). Testing whether an algorithm achieves approximate fairness (under relaxed fairness constraints) while maintaining sufficient overall utility has become a question of central importance.

Motivated by this challenge, our paper proposes a statistical test that jointly evaluates group fairness and utility, which forms the main focus of our study.

## 1.1 Overview of the Utility-Constrained Fairness Testing Framework

Our statistical hypothesis testing framework enables auditors to determine whether the utility-constrained biases observed in an audit reflect inherent properties of the algorithm or simply arise from randomness in the data. The framework is also designed to function as a black-box, requiring no knowledge of the internal structure of the algorithm. The framework adopts a relaxed version of the *strong demographic parity* (SDP) notion (Jiang et al., 2020) to evaluate approximate fairness (see Section 2.3) and incorporates a utility function inspired by the potential outcome framework in causal inference (Rubin, 2005) (see Section 2.1).

We adopt the potential outcomes framework to define overall utility (Imbens & Rubin, 2015). Specifically, we consider a two-level treatment $W_i \in \{0, 1\}$ and an outcome $Y_i \in \mathbb{R}$, interpreted as utility, where each individual's utility is determined by the treatment, non-sensitive context, and sensitive context. To reflect the utility trade-off, the auditor needs to ensure that the overall expected utility $\mathbb{E}[Y_i(W_i)]$ exceeds a specified threshold (see Section 3 for details). While our analysis focuses on binary treatments and binary sensitive attributes, the results naturally extend to multi-level or continuous treatments and multiple sensitive attributes using similar proof techniques. For clarity and readability, we confine our discussion to the binary case and discuss the extensions in Appendix C.2.

For fairness evaluation, a commonly used criterion is statistical parity (SP) (Agarwal et al., 2019)—also referred to as demographic parity (DP) (Dwork et al., 2012) or disparate impact (Feldman et al., 2015)—which requires statistical independence between classifier predictions and sensitive attributes. However, as noted by Jiang et al. (2020), SP/DP has important practical limitations: it is highly sensitive to threshold choices, meaning that satisfying the criterion at one threshold does not guarantee that it holds at others (see Section 2.3 for details). To address this issue, Jiang et al. (2020) has proposed the fairness criterion of *strong demographic parity* (SDP), which requires that decisions be independent of sensitive attributes across all thresholds. Building on this idea, we formalize a relaxed version of SDP within a utility-constrained testing framework (see Definition 3). We evaluate whether the propensity score aligns with the approximate SDP fairness criterion in our framework. Beyond the specific fairness notion and utility definition considered here, our hypothesis testing framework can be readily extended to other formulations of utility-constrained fairness. Details are provided in Appendix C.1.

Our hypothesis testing framework addresses the statistical difficulties that stem from simultaneously accounting for multiple criteria — fairness and utility trade-offs — through the use of Wasserstein projection techniques. In essence, the test statistic is obtained by optimally transporting the empirical distribution onto the class of probability models that satisfy the specified group fairness requirements. In this way, we evaluate whether the utility-constrained fairness criterion is plausibly satisfied under the true data-generating process. The hypothesis is rejected if the computed test statistic exceeds a critical value determined by the chosen significance level. This critical value is obtained from the asymptotic behavior of the test statistic, which forms one of the main results of this work.

We summarize our main contributions as follows. (1) To the best of our knowledge, our paper is the first to implement a hypothesis test of fairness under a fairness–utility trade-off framework using the Wasserstein projection method. (2) The proposed test is computationally tractable, interpretable, and broadly applicable to a wide range of machine learning and AI algorithms used for estimating propensity scores and outcome models. (3) Our framework is readily extendable beyond the specific fairness and utility criteria considered here, opening avenues for future research. (4) We empirically illustrate the application of our hypothesis test framework to real-world data.

## 1.2 Related Work

The field of algorithmic fairness has expanded rapidly, yielding numerous definitions and approaches. Early work focused on demographic parity (also known as statistical parity or disparate impact) (Calders et al., 2013; Feldman et al., 2015; Zafar et al., 2017), requiring equal decision

probabilities across groups; equalized odds (Hardt et al., 2016), requiring false positive and false negative rates to be independent of group membership; and equal opportunity along with its probabilistic variants (Hardt et al., 2016; Pleiss et al., 2017), aimed at reducing disparities in favorable outcomes. Yet no single definition has emerged as standard, and — beyond trivial cases — no algorithm can satisfy multiple criteria simultaneously. For comprehensive surveys, see (Pessach & Shmueli, 2022; Chen et al., 2024).

Our study also connects to the body of work on fairness–utility trade-offs (Corbett-Davies et al., 2017). A central observation in this literature is that unconstrained predictors typically achieve utility that is at least as high as, and often higher than, predictors subject to fairness constraints. Numerous studies document utility losses when fairness constraints are imposed (Mitchell et al., 2021), and propose strategies to manage this trade-off (Fish et al., 2016). Still, the existence and magnitude of such trade-offs remain divided. For example, Rodolfa et al. (2021) reports that fairness–utility trade-offs are minimal in practice, while others contend that such trade-offs may not exist (Maity et al., 2020; Dutta et al., 2020). The impact ultimately depends on the specific fairness definition under consideration, with studies downplaying trade-offs often focusing on criteria like equalized odds (Hardt et al., 2016) or (multi-)calibration (Chouldechova, 2017), which differ from the fairness notions examined in our work.

We ground our notion of utility in the potential outcomes framework from causal inference (Rubin, 2005; Imbens & Rubin, 2015), which naturally links our work to the causal fairness literature. Yet, this literature has paid comparatively little attention to the trade-off between fairness and utility. Notable exceptions include Nilforoshan et al. (2022), who demonstrate that for any policy satisfying a causal fairness constraint, one can typically construct an alternative policy with strictly higher utility and the same total variation (TV) distance; and Plecko & Bareinboim (2024), who analyze decision scores used in policy design and show how disparities in these scores may affect utility. Recently, Plecko & Bareinboim (2025) has introduced a systematic framework for analyzing the fairness–accuracy trade-off from a causal fairness perspective, showing that such trade-offs almost always arise.

Methodologically, our hypothesis testing framework connects to the literature on statistical inference using projection-based criteria (Owen, 2001; Blanchet et al., 2019; Cisneros-Velarde et al., 2020). Our approach is also related to Taskesen et al. (2021) and Si et al. (2021), who cast fairness questions as hypothesis testing problems using the Robust Wasserstein Profile Inference method of Blanchet et al. (2019). Whereas Taskesen et al. (2021) and Si et al. (2021) examine specific fairness notions imposed as hard or relaxed constraints — without parameters to capture utility trade-offs — our framework is designed for settings in which such trade-offs are explicitly modeled.

**Notations.** Given a measurable set $\mathcal{Z} \subset \mathbb{R}^d$, we use $\mathcal{P}(\mathcal{Z})$ to denote the set of probability distributions on $\mathcal{Z}$ that are square integrable. For a sequence $\{\xi_n\}_{n \geq 1}$, we say $\xi_n \Rightarrow \xi$ means $\xi_n$ converges in probability to $\xi$. $\|\cdot\|$ denotes the Euclidean norm on $\mathbb{R}^d$. For two random variables $X, Y$, $X \stackrel{d}{=} Y$ means $X, Y$ follow the same distribution, and $X \perp\!\!\!\perp Y$ means $X$ is independent of $Y$. We use $\mathbb{P}(\cdot)$ to denote the general probability measure (unless specified otherwise), $\mathbb{E}[\cdot]$ as the expectation, and $\mathbf{1}\{\cdot\}$ as the indicator function. $\mathrm{Unif}[0,1]$ denotes the uniform distribution over $[0,1]$. $\iff$ means "if and only if". Given a matrix or vector $A$, $A^T$ means the transpose of $A$. We use $\mathcal{N}(\boldsymbol{\mu}, \boldsymbol{\Sigma})$ as the Gaussian distribution with mean $\boldsymbol{\mu}$ and covariance $\boldsymbol{\Sigma}$. Given a random variable $X$ and a distribution $\mathcal{F}$, $X \sim \mathcal{F}$ means that $X$ follows $\mathcal{F}$. Given a subset $\mathcal{Z} \subset \mathbb{R}^d$, for any function $f : \mathcal{Z} \to \mathbb{R}$, we use $Df(\cdot)$ to denote the gradient of $f$. We use $\infty$ to denote infinity, and $\delta_x$ to denote the delta measure that puts all probability mass on a single point $x$.

## 2 PROBLEM SETUP AND PRELIMINARIES

The outcome space is given by $\mathcal{Y} \subset \mathbb{R}$, the covariate space is given by $\mathcal{X} \subset \mathbb{R}^d$, while the sensitive attribute space is $\mathcal{S} = \{0, 1\}$. We consider random variables $\{(Y_i, X_i, S_i, W_i)\}_{i=1}^N$ that are drawn *independently and identically distributed* (i.i.d.) from a fixed but unknown distribution. $X_i$ represents the non-sensitive covariates, $S_i \in \{0, 1\}$ denotes a sensitive attribute such as gender or race, and $W_i \in \{0, 1\}$ denotes the binary treatment level. Our setting follows the potential outcome framework (Imbens & Rubin, 2015). The observed outcome is $Y_i = Y_i(W_i)$, which corresponds to the realized utility, whereas the counterfactual outcome $Y_i(1 - W_i)$ is unobserved. We refer

to $W_i = 1$ as individual $i$ receiving the treatment, and $W_i = 0$ as receiving the control. Denote $\pi_a(x) := \mathbb{P}(W_i = 1 \mid X_i = x, S_i = a)$ as the probability that individual $i$ receives the treatment given contexts $(X_i, S_i) = (x, a)$, where $\pi_a : \mathcal{X} \to [0, 1]$ for $a \in \{0, 1\}$. We refer to $\pi_a(x)$ as the propensity score for context $(x, a)$ throughout the paper. Specifically, on observing the realization of context $\{X_i = x, S_i = a\}$ for individual $i$, the decision maker randomly assigns a treatment level $W_i \in \{0, 1\}$ according to the propensity score $\pi_a(x)$, after which the corresponding utility $Y_i(W_i)$ is observed. Figure 1 shows how $(X, S)$ influence $W$ and $(X, S, W)$ determine $Y$. Although we focus on binary treatment levels and binary sensitive attributes, the results readily extend to multi-level or continuous treatments and multiple sensitive attributes, with similar proof techniques. For clarity and readability, we restrict attention to the binary case, and discuss the extensions in Appendix C.2.

## 2.1 UTILITY

For any $w \in \{0, 1\}$, we denote $m_w(x, a) := \mathbb{E}[Y_i(w)|X_i = x, S_i = a]$ as the expected utility of treatment level $w$ for the population with non-sensitive covariate $x$ and sensitive attribute $a$. Denote $p_a(x) := \mathbb{P}(S_i = a|X_i = x)$ for any $a \in \{0, 1\}$. We impose the following assumption:

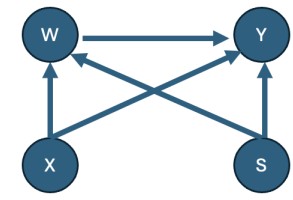

Figure 1: Graphical illustration: $(X, S)$ influence $W$ and $(X, S, W)$ influence $Y$.

**Assumption 1.** *Unconfoundedness:* $W_i \perp\!\!\!\perp \{Y_i(1), Y_i(0)\}|X_i, S_i$. *(ii) Boundedness:* $0 \le Y_i(1), Y_i(0) \le B$ *for some bounded constant* $B > 0$.

By definition, the expected utility is equal to

$$\mathbb{E}\left[Y_i(W_i)\right] =_{(a)} \mathbb{E}\left[W_i Y_i(1) + (1 - W_i)Y_i(0)\right] =_{(b)} \mathbb{E}\left[\mathbb{E}\left[Y_i(1)W_i + (1 - W_i)Y_i(0)|X_i, S_i\right]\right]$$

$$=_{(c)} \mathbb{E}[m_1(X_i, S_i)\pi_{S_i}(X_i) + m_0(X, S_i)(1 - \pi_{S_i}(X_i))]$$

$$=_{(d)} \sum_{a \in \mathcal{S}} \mathbb{E}\left[\{m_1(X_i, a)\pi_a(X) + m_0(X_i, a)(1 - \pi_a(X_i))\} p_a(X_i)\right].$$

$$(1)$$

where in (1), (a) follows from the definition of the potential outcomes, (b) uses tower property, (c) follows from (i) of Assumption 1. Although Assumption 1 is standard in the literature, it may not always hold in practice — particularly the unconfoundedness condition. To address this in practice, we verify in Appendix E that Assumption 1 holds in our empirical studies with real data.

## 2.2 OPTIMAL TRANSPORT AND WASSERSTEIN DISTANCE

Let $\mathcal{P}(\mathcal{X})$ denote the set of all probability distributions on $\mathcal{X}$. According to (d) of (1), the expected utility can be expressed as the expectation of a function of $X_i$, with the expectation taken with respect to the distribution of $X_i$. We now introduce the notion of optimal transport costs via Wasserstein distance:

**Definition 1** (Optimal transport costs and Wasserstein Distance). *Given a lower semicontinuous function* $c : \mathcal{X} \times \mathcal{X} \to [0, \infty]$*, the type-2 Wasserstein optimal transport cost* $\mathcal{W}_c(\mathbb{Q}_1, \mathbb{Q}_2)$ *for any* $\mathbb{Q}_1, \mathbb{Q}_2 \in \mathcal{P}(\mathcal{X})$ *is defined as*

$$\mathcal{W}_c(\mathbb{Q}_1, \mathbb{Q}_2) = \min_{\pi \in \Gamma(\mathbb{Q}_1, \mathbb{Q}_2)} \sqrt{\mathbb{E}_\pi[c(X, X')^2]},$$

*where* $\Gamma(\mathbb{Q}_1, \mathbb{Q}_2)$ *is the set of all joint distributions of* $(X, X')$ *such that the distribution of* $X$ *is* $\mathbb{Q}_1$ *and the distribution of* $X'$ *is* $\mathbb{Q}_2$.

When $c(\cdot, \cdot)$ is a metric on $\mathcal{X}$, $\mathcal{W}_c(\cdot, \cdot)$ is the Wasserstein distance Villani et al. (2009). Note that in the existing literature on testing fairness via Wasserstein projection, the focus is on computing Wasserstein distances between distributions on $\mathcal{X} \times \mathcal{S} \times \mathcal{Y}$ (Taskesen et al., 2021; Si et al., 2021). The ground metric is typically defined as

$$c((x, a, y), (x', a', y')) = \|x - x'\| + \infty\|a - a'\| + \infty\|y - y'\|,$$

where $\| \cdot \|$ is a norm on $\mathbb{R}^d$. This formulation assumes absolute trust in the sensitive attribute and outcome observed in the training data. Consequently, the transport cost depends only on the distribution of $X_i$. Such an absolute-trust restriction is standard in the fair machine learning literature (Xue et al., 2020; Taskesen et al., 2020). Hence, we follow same absolute-trust assumption and restrict attention to optimal transport over distributions in $\mathcal{P}(\mathcal{X})$.

Conceptually, the Wasserstein distance captures not only pointwise differences between distributions but also the cost of rearranging their probability mass. This makes the Wasserstein framework a powerful tool for comparing complex distributions while preserving geometric information about $\mathcal{X}$. Such a perspective is particularly valuable in fairness applications, where aligning group distributions is often a key goal, and the optimal transport view provides a direct way to assess how populations overlap or diverge in the covariate space $\mathcal{X}$.

### 2.3 APPROXIMATE STRONG DEMOGRAPHIC PARITY

As noted in the introduction, achieving absolute fairness is nearly always impossible once utility trade-offs are taken into account. Thus, rather than adopting fairness notions that impose strict criteria, we propose a relaxed fairness definition inspired by the *Strong Demographic Parity* (SDP) criterion introduced by Jiang et al. (2020). Firstly, the notion of SDP is defined as:

**Definition 2** (Strong Demographic Parity). *We say that SDP is satisfied if* $\pi_{S_i}(X_i) \perp\!\!\!\perp S_i$.

Jiang et al. (2020) introduce the notion of Strong Demographic Parity (SDP) from the perspective of a binary classifier. In their setting, $W_i$ is the binary label, $X_i$ and $S_i$ denote the non-sensitive and sensitive features, and $\mathbb{P}(W_i = 1 \mid X_i, S_i) \in [0, 1]$ represents the model's predicted probability that unit $i$ belongs to class 1. A class prediction $\hat{W}_i \in \{0, 1\}$ is then obtained via a threshold rule $\tau \in [0, 1]$, with $\hat{W}_i := \mathbf{1}\{\mathbb{P}(W_i = 1 \mid X_i, S_i) > \tau\}$. The standard *demographic parity* (DP) criterion requires $\mathbb{P}(\hat{W}_i = 1|S_i = 1) = \mathbb{P}(\hat{W}_i = 1|S_i = 0)$, but satisfying DP at one threshold does not guarantee that it holds for others. To address this limitation, SDP requires $\pi_{S_i}(X_i) \perp\!\!\!\perp S_i$, ensuring independence from the sensitive attribute across all thresholds. Moreover, SDP implies DP for every possible threshold $\tau$.

In our setting, let $p_{\pi_a(X_i)}$ denotes the probability density function (pdf) of $\pi_a(X_i)$ for $a \in \{0, 1\}$. So SDP can also be defined as $p_{\pi_1(X_i)} = p_{\pi_0(X_i)}$, which holds if and only if

$$\mathbb{E}_{\tau \sim \text{Unif}[0,1]} \left[ |\mathbb{Q}(\pi_1(X_i) > \tau) - \mathbb{Q}(\pi_0(X_i) > \tau)| \right] = 0, \tag{2}$$

where $\mathbb{Q}$ is the distribution of $X_i$. Indeed, let $\mathcal{W}_1$ be the 1-Wasserstein distance (i.e. setting $c(x, x') = |x - x'|$ in Definition 1), by Lemma B.1, (2) holds $\iff \mathcal{W}_1(\pi_1(X_i), \pi_0(X_i)) = 0 \iff p_{\pi_1(X_i)} = p_{\pi_0(X_i)}$. Now we define a relaxed fairness concept based upon SDP:

**Definition 3** ($\epsilon$-Approximate SDP). *We say that $\epsilon$-approximate SDP is satisfied if*

$$\mathbb{E}_{\tau \sim \text{Unif}[0,1]} \left[ |\mathbb{Q}(\pi_1(X_i) > \tau) - \mathbb{Q}(\pi_0(X_i) > \tau)| \right] \leq \epsilon.$$

In practice, practitioners may tune the parameter $\epsilon$ to meet application-specific needs. Si et al. (2021) also adopts a related idea of fairness relaxation in their extended framework, but the fairness notion they consider differs substantially from ours.

## 3 TESTING UTILITY-CONSTRAINED FAIRNESS VIA OPTIMAL TRANSPORT

Denote $\mathcal{Z} = \mathcal{X} \times \{0, 1\} \times \mathcal{S} \times \mathcal{Y}$ as the space where the random vector $(X_i, W_i, S_i, Y_i)$ is supported on. Recall that $\mathcal{P}(\mathcal{Z})$ is the set of probability distributions on $\mathcal{Z}$. Given $\epsilon \geq 0$, $r \in \mathbb{R}$, we define

$$\mathcal{G}(r, \epsilon) := \left\{ \widetilde{\mathbb{Q}} \in \mathcal{P}(\mathcal{Z}) \,\middle|\, \begin{array}{c} \mathbb{E}_{\widetilde{\mathbb{Q}}}[Y_i(W_i)] \geq r \\ \mathbb{E}_{\tau \sim \text{Unif}[0,1]} \left[ \left| \widetilde{\mathbb{Q}}_X(\pi_1(X_i) > \tau) - \widetilde{\mathbb{Q}}_X(\pi_0(X_i) > \tau) \right| \right] \leq \epsilon \end{array} \right\}, \tag{3}$$

where $\widetilde{\mathbb{Q}}_X$ is the marginal distribution of $X_i$ (obtained by integrating $\widetilde{\mathbb{Q}}$ with respect to the marginals of $(W_i, S_i, Y_i)$). Formally, $\mathcal{G}(r, \epsilon)$ is defined as the set of joint distributions of $(X_i, W_i, S_i, Y_i)$ that satisfy $\epsilon$-approximate SDP and guarantee an overall expected utility of at least $r$. Given $N$ samples

$\{x_i, w_i, s_i, y_i\}_{i \in [N]}$ drawn i.i.d. from a distribution $\widetilde{\mathbb{P}}$ of $(X_i, W_i, S_i, Y_i)$, we are interested in the statistical test with the composite null hypothesis:

$$\mathcal{H}_0 : \widetilde{\mathbb{P}} \in \mathcal{G}(r, \epsilon) \quad \text{v.s.} \quad \mathcal{H}_1 : \widetilde{\mathbb{P}} \notin \mathcal{G}(r, \epsilon). \tag{4}$$

Define

$$\mathcal{F}_{r,\epsilon} := \left\{ \mathbb{Q} \in \mathcal{P}(\mathcal{X}) \middle| \begin{array}{l} \sum_{a \in \mathcal{S}} \mathbb{E}_{\mathbb{Q}} \left[ \{m_1(X_i, a)\pi_a(X) + m_0(X_i, a)(1 - \pi_a(X_i))\} p_a(X_i) \right] \geq r \\ \\ \mathbb{E}_{\tau \sim \mathrm{Unif}[0,1]} \left[ |\mathbb{Q}(\pi_1(X_i) > \tau) - \mathbb{Q}(\pi_0(X_i) > \tau)| \right] \leq \epsilon. \end{array} \right\}, \tag{5}$$

Recall from (d) of (1) that $\mathbb{E}_{\widetilde{\mathbb{Q}}}[Y_i(W_i)] \geq r$ is equivalent to

$$\sum_{a \in \mathcal{S}} \mathbb{E}_{\widetilde{\mathbb{Q}}_X} \left[ \{m_1(X_i, a)\pi_a(X) + m_0(X_i, a)(1 - \pi_a(X_i))\} p_a(X_i) \right] \geq r, \tag{6}$$

So given that $X_i \sim \mathbb{P}$, testing (4) is equivalent to the following hypothesis test:

$$\mathcal{H}_0 : \mathbb{P} \in \mathcal{F}_{r,\epsilon} \quad \text{v.s.} \quad \mathcal{H}_1 : \mathbb{P} \notin \mathcal{F}_{r,\epsilon}. \tag{7}$$

In other words, testing the null hypothesis (4) for the joint distribution of $(X_i, W_i, S_i, Y_i)$ reduces to testing the corresponding hypothesis for the marginal distribution of $X_i$, given that we have an absolute trust in the training sample, and that unconfoundedness holds according to Assumption 1.

In order to propose a proper test statistic, we denote $\hat{\mathbb{P}}_N = N^{-1} \sum_{i=1}^{N} \delta_{x_i}$ as the empirical measure of the samples obtained from a distribution $\mathbb{P} \in \mathcal{P}(\mathcal{X})$. The projection distance of $\hat{\mathbb{P}}_N$ unto $\mathcal{F}_{r,\epsilon}$ is defined as

$$
\begin{aligned}
\mathcal{R}_{r,\epsilon}(\hat{\mathbb{P}}_N) \quad &:= \quad \inf_{\mathbb{Q} \in \mathcal{F}_{r,\epsilon}} \mathcal{W}_c(\mathbb{Q}, \hat{\mathbb{P}}_N)^2 \\
&= \left\{ \begin{array}{l} \inf_{\mathbb{Q} \in \mathcal{P}(\mathcal{X})} \mathcal{W}_c(\mathbb{Q}, \hat{\mathbb{P}}_N)^2 \\ \\ \text{s.t.} \quad \sum_{a \in \mathcal{S}} \mathbb{E}_{\mathbb{Q}}[\{m_1(X, a)\pi_a(X) + m_0(X, a)(1 - \pi_a(X))\} p_a(X)] \geq r \\ \\ \mathbb{E}_{\tau \sim \mathrm{Unif}[0,1]} \left[ |\mathbb{Q}(\pi_1(X) > \tau) - \mathbb{Q}(\pi_0(X) > \tau)| \right] \leq \epsilon \end{array} \right\}
\end{aligned}
\tag{P}
$$

When $\epsilon = 0$ and $r = -\infty$, (P) corresponds to testing the strict strong demographic parity without considering any utility tradeoff. As $r$ increases and $\epsilon$ decreases, the constraints become more stringent, and for some $(\epsilon, r)$ no probability measure may satisfy (P). Similar trade-offs have been observed empirically in prior work under alternative fairness metrics and related perspectives (Plecko & Bareinboim, 2025; Maity et al., 2020; Dutta et al., 2020, etc.). The choice of $(\epsilon, r)$ naturally depends on the empirical context under study. For example, in a consumer lending setting, the decision maker may require that expected repayment (or profit) remains above a threshold $r$, while $\epsilon$ controls the tolerated disparity in loan approval rates between minority and majority groups across all classification thresholds. In contrast, in a healthcare intervention scenario, $r$ could represent the minimum expected improvement in patient outcomes (e.g., reduction in hospitalization rates), whereas $\epsilon$ governs the allowable imbalance in treatment assignment probabilities across genders. These examples illustrate how $(\epsilon, r)$ jointly capture the trade-off between maintaining sufficient utility and ensuring fairness across sensitive groups.

For a given significance level $\alpha$ and $\eta_{1-\alpha}$ as the $(1-\alpha)$ quantile of some limiting distribution related to the test statistic $t_N$, we reject the hypothesis $\mathcal{H}_0$ if $t_N > \eta_{1-\alpha}$. For the remainder of the paper, we set $c(x, x') = \|x - x'\|$ in Definition 1, where $\|\cdot\|$ denotes the Euclidean norm on $\mathbb{R}^d$.

### 3.1 STRONG DUALITY

We provide the following additional regularity assumptions:

**Assumption 2.** $m_1(\cdot, a)$, $m_0(\cdot, a)$, $\pi_a(\cdot)$ *are continuously differentiable with derivatives* $Dm_1(\cdot, a)$, $Dm_0(\cdot, a)$ *and* $D\pi_a(\cdot)$ *for* $a \in \{0, 1\}$.

**Assumption 3.** *There exists some $x \in \mathcal{X}$, such that $\pi_1(x) = \pi_0(x)$ and*

$$\sum_{a \in \{0,1\}} p_a(x)[m_1(x,a)\pi_a(x) + m_0(x,a)(1 - \pi_a(x))] \geq r.$$

Assumption 3 posits that the expected utility attains the reservation level $r$ for some covariate. This condition is essential; without it, no distribution of the covariate $X$ could yield an overall expected utility of $r$, rendering the framework incoherent.

We now present the first main result of the paper, a strong duality result for the projection distance defined by (P):

**Theorem 3.1** (Strong Duality). *Under Assumptions 1, 2, 3, we have*

$$\mathcal{R}_{r,\epsilon}(\hat{\mathbb{P}}_N) = \sup_{(\lambda,\alpha) \in \mathbb{R}_+ \times \mathbb{R}_+} \lambda r - \alpha \epsilon$$

$$+ \frac{1}{N} \sum_{i=1}^{N} \min_{x \in \mathcal{X}} \{\|x - X_i\|^2 + \alpha|\pi_1(x) - \pi_0(x)| - \lambda M(x)\},$$

*where $M(x) = \sum_{a \in \{0,1\}} \{m_1(x,a)\pi_a(x) + m_0(x,a)(1 - \pi_a(x))\}p_a(x)$.*

### 3.2 ASYMPTOTICS FOR THE PROJECTION DISTANCE

We now study the limiting behavior of the projection distance $\mathcal{R}_{r,\epsilon}(\hat{\mathbb{P}}_N)$. Define

$$V_+ := (DM(X_i)^T[D(\pi_1 - \pi_0)(X_i)], -\|D(\pi_1 - \pi_0)(X_i)\|^2),$$
$$V_- := (DM(X_i)^T[D(\pi_1 - \pi_0)(X_i)], \|D(\pi_1 - \pi_0)(X_i)\|^2),$$
$$S_+ := \begin{pmatrix} DM(X_i) \\ -D[\pi_1 - \pi_0](X_i) \end{pmatrix}, \quad S_- := \begin{pmatrix} DM(X_i) \\ D[\pi_1 - \pi_0](X_i) \end{pmatrix}.$$

For $\zeta \in \mathbb{R}^2$ and given vector $w \in \mathbb{R}^2$, define $f^+(\zeta) := \max\{2\mathbb{E}\left[S_+ S_+^T \mathbf{1}\{\zeta^T V_+ \geq 0\}\right]^{-1} w, 0\}$, $f^-(\zeta) := \max\left\{2\mathbb{E}\left[S_- S_-^T \mathbf{1}\{\zeta^T V_- < 0\}\right]^{-1} w, 0\right\}$. We impose the following regularity condition:

**Assumption 4.** $f^+$, $f^-$ *both have fixed points.*

Note that we allow $w \in \mathbb{R}^2$ to be arbitrary, so Banach's fixed-point theorem based on the contraction condition does not directly apply for fixed-point results. To verify Assumption 4, we may adopt the results from several extensions of the contraction principle that have been developed in the literature (Boyd & Wong, 1969; Caristi, 1979; Bessaga, 1959); see Pata et al. (2019) for a comprehensive review.

We now present the second main result of this section for the asymptotic behavior of the projection distance. For a sequence of random events $A_N$, we write $A_N \lesssim_D B$ if, for every bounded, continuous, and nondecreasing function $g$, $\limsup_{N \to \infty} \mathbb{E}[g(A_N)] \leq \mathbb{E}[g(B)]$.

**Theorem 3.2** (Stochastic Upper Bound). *Suppose Assumptions 1, 2, 3, 4 hold. Then under the null hypothesis $\mathcal{H}_0$,*

$$N\mathcal{R}_{r,\epsilon}(\hat{\mathbb{P}}_N) \lesssim_D \max \left\{ \begin{array}{c} \overline{W}^T \mathbb{E}\left[S_+ S_+^T \mathbf{1}\{\zeta_+^{*T} V_+ \geq 0\}\right]^{-1} \overline{W}, \\ \overline{W}^T \mathbb{E}\left[S_- S_-^T \mathbf{1}\{\zeta_-^{*T} V_- \geq 0\}\right]^{-1} \overline{W} \end{array} \right\} \mathbf{1}\{\overline{W} \geq 0\}, \quad (8)$$

*where $\overline{W} = \begin{pmatrix} \overline{M} \\ \overline{\Pi} \end{pmatrix}$, $\overline{M} \sim \mathcal{N}(0, \text{cov}[M(X_i)])$, $\overline{\Pi} \sim \mathcal{N}(0, \text{cov}[|\pi_1(X_i) - \pi_0(X_i)|])$, and*

$$\zeta_+^* = \max\left\{2\mathbb{E}\left[S_+ S_+^T \mathbf{1}\{\zeta_+^{*T} V_+ \geq 0\}\right]^{-1} \overline{W}, 0\right\}, \quad (9)$$

$$\zeta_-^* = \max\left\{2\mathbb{E}\left[S_- S_-^T \mathbf{1}\{\zeta_-^{*T} V_- < 0\}\right]^{-1} \overline{W}, 0\right\}. \quad (10)$$

Theorem 3.2 implies that we can use $t_N(\epsilon, r) = N\mathcal{R}_{r,\epsilon}(\hat{\mathbb{P}}_N)$ as a test statistic, leveraging the stochastic upper bound established in Theorem 3.2. Given a significance level $\alpha$, let $\eta_{1-\alpha}$ be the $(1 - \alpha)$ quantile of the right hand side of (8). Following the hypothesis testing framework proposed according to (7) and (P), we reject $\mathcal{H}_0$ if $t_N(\epsilon, r) > \eta_{1-\alpha}$, which results in a conservative test and the type I error is less than or equal to $\alpha$ asymptotically.

### 3.3 COMPUTATIONS

To compute the test statistic $N\mathcal{R}_{r,\epsilon}(\hat{\mathbb{P}}_N)$, recall that $\mathcal{R}_{r,\epsilon}(\hat{\mathbb{P}}_N)$ is defined by (P):

$$\mathcal{R}_{r,\epsilon}(\hat{\mathbb{P}}_N) = \begin{cases} \sup & \lambda r - \alpha\epsilon + \frac{1}{N}\sum_{i=1}^{N}\gamma_i(\lambda,\alpha) \\ \text{s.t.} & \lambda \geq 0, \alpha \geq 0 \end{cases} \tag{11}$$

and $\gamma_i(\lambda,\alpha) := \min_{x\in\mathcal{X}}\{\|x-X_i\|^2 + \alpha|\pi_1(x)-\pi_0(x)| - \lambda M(x)\}$. Note that $\|x-X_i\|^2 + \alpha|\pi_1(x)-\pi_0(x)| - \lambda M(x)$ is concave in $\alpha, \lambda$ for any $x \in \mathcal{X}$, and that the minimum of a family of concave function is still concave, so $\gamma_i(\lambda,\alpha)$ is concave $\forall i \in [n]$. If minimizing $\|x-X_i\|^2 + \alpha|\pi_1(x)-\pi_0(x)| - \lambda M(x)$ over $x \in \mathcal{X}$ can be solved easily for any $\lambda \geq 0, \alpha \geq 0$, then the computation is straightforward. For example, we may require $M(\cdot)$ to be concave and $\pi_1(x)-\pi_0(x)$ to be affine in $x$, so that the objective $\|x-X_i\|^2 + \alpha|\pi_1(x)-\pi_0(x)| - \lambda M(x)$ is convex in $x$. For general algorithms addressing non-convex optimization problems, we refer to the methods developed in Allen-Zhu & Hazan (2016); Jain et al. (2017); Danilova et al. (2022); Chen et al. (2018); Dauphin et al. (2014).

We proceed as follows to compute the quantile of the stochastic upper bound given on the right-hand side of equation 8: (i) compute $\zeta_+^*$, $\zeta_-^*$ defined by (9) and (10) via iterative methods. (ii) Compute the inverse matrices $\mathbb{E}\left[S_+S_+^T\mathbf{1}\{\zeta_+^{*T}V_+ \geq 0\}\right]^{-1}$ and $\mathbb{E}\left[S_-S_-^T\mathbf{1}\{\zeta_-^{*T}V_- \geq 0\}\right]^{-1}$ by approximating $\mathbb{E}\left[S_+S_+^T\mathbf{1}\{\zeta_+^{*T}V_+ \geq 0\}\right]$ and $\mathbb{E}\left[S_-S_-^T\mathbf{1}\{\zeta_-^{*T}V_- \geq 0\}\right]$ via sample average approximations or weighted sample average. (iii) Draw samples of $\overline{W}$ defined as in Theorem 3.2 and compute the quantile via standard bootstrap method.

## 4 NUMERICAL EXPERIMENTS

We first implement our hypothesis test framework in a case study of a synthetic pricing problem between elder and young buyers (Kahneman & Tversky, 2013), then conduct experiments on three real datasets with sensitive attributes and show the fairness-accuracy trade-off of Tikhonov-regularized logistic classifiers and SVM classifiers.

**Simulated Data: Pricing Policies.** We first consider non-sensitive click-rate information denoted by $x \in [0,1]$, which follows uniform distributions. Meanwhile, the sensitive attribute—customer age—is represented by a binary variable $a \in \{0,1\}$, distinguishing between different demographic groups. Additionally, the treatment variable $w \in \{0,1\}$ indicates the treatment level applied to each individual. The $a = 0$ category represents elder buyers with stable preferences, favoring predictable treatments $w = 0$, and the $a = 1$ category corresponds to young buyers, who are more risk-taking and price-sensitive, favoring volatile treatments $w = 1$. The propensity score is defined as $\pi_a(x) = \theta_a x$ where $0 \leq \theta_a \leq 1$ and $a \in \{0,1\}$. The conditional expected utility function is $m_w(x,a) = \beta_0^{(a)} + \beta_1^{(a)}w + \beta_2^{(a)}x$, where $(\beta_0^{(0)}, \beta_1^{(0)}, \beta_2^{(0)}) = (0.8, 0.5, 0.7)$ for elder buyers $(a = 0)$ and $(\beta_0^{(1)}, \beta_1^{(1)}, \beta_2^{(1)}) = (0.5, 1.0, 0.5)$ for young buyers $(a = 1)$. We implement the hypothesis test for the policies parametrized by $\theta_1 \in (0.55, 0.6, 0.65, 0.7, 0.75, 0.8, 0.85, 0.9)$ and $\theta_0 = 1 - \theta_1$. By definition, Assumption 1 follows directly.

Figure 2 illustrates the trade-off between utility and fairness for $r = 0.8, 0.9, 1.0, 1.1, 1.2$ and $1.3$ with fixed $\epsilon = 0.01$. As the utility requirement becomes more stringent (larger $r$), the test statistic (blue curve) increases substantially, while the stochastic upper bound at significance level $\alpha = 0.05$ decreases. The figure also demonstrates the impact of varying $\epsilon$ values ($\epsilon = 0.05, 0.10, 0.15, 0.20, 0.25, 0.30$) for approximate fairness criteria defined via $\epsilon$-approximate SDP. The results indicate that as the fairness criterion is relaxed (i.e., as $\epsilon$ increases), the policy is deemed fairer, and the level-0.05 test is rejected at larger values of $\theta_1$.

**Empirical Study** In this experiment, we evaluate the fairness of binary classifiers under varying regularization weights. We use three typical datasets with sensitive attribute information: COMPAS (Dua et al., 2017), Arrhythmia (Angwin et al., 2016) and Drug (Fehrman et al., 2017). The details of the datasets are provided in Appendix E. The policies of COMPAS and Arrhythmia datasets are modeled via Tikhonov-regularized logistic regression and the policies of Drug dataset are modeled via naive SVM classifiers parametrized by the ridge regularization. We estimate $M(x)$ using a Gaussian kernel-based method. Figure 4 in Appendix F presents the test statistics, fairness rejection

Figure 2: Simulated data results. Test statistics and confidence intervals.

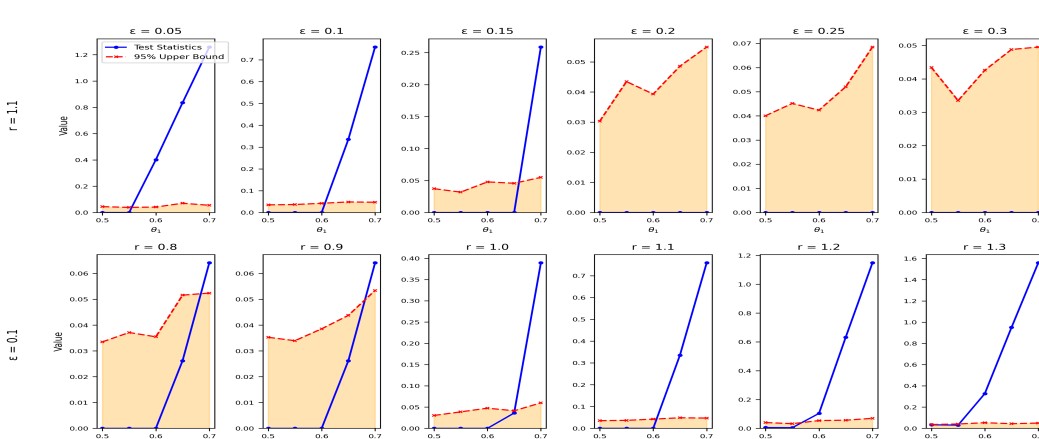

threshold, and classifier accuracy of the three datasets. A clear trade-off emerges between model accuracy and approximate fairness metrics as the regularization factor is adjusted.

**Comparison with Naive Bootstrap Tests** We also compare our hypothesis test framework with the naive bootstrap baseline for different values of $(\epsilon, r)$. For the naive bootstrap baseline, it rejects whenever either of the constraints of equation P fails in the bootstrap sample (i.e. either overall utility is too low or $\epsilon$-SDP constraint fails). We compute the fairness hypothesis rejection rates for different values of $(\epsilon, r)$ under level-0.05 tests. Table 2 and Figure 3 in Appendix F summarize empirical study results. Figure 5 in Appendix F summarizes simulated data results. We find that bootstrap is consistently more conservative than our method: The bootstrap test is consistently more likely to reject the fairness hypothesis than our test across all values of $(\epsilon, r)$. Our test statistic is a Wasserstein projection distance that jointly enforces both the utility and relaxed SDP constraints. This projection formulation captures the interaction between the two conditions and the non-smooth boundary of the feasible set. In contrast, a simple bootstrap treats the two quantities as independent scalars and ignores the geometry induced by the projection. Recall that Theorem 3.2 establishes that our test is conservative, as it is constructed using a stochastic upper bound. The naive bootstrap is even more conservative. These observations further underscore the validity and usefulness of our proposed method.

**Remark 1.** *Here we comment that Assumption 1 holds for all empirical illustrations:* **Uncon-foundedness:** *In our experimental framework, all treatments are derived from Tikhonov-regularized Logistic Regression and SVM classifiers. Since these models' predictions depend solely on the input features $(x, a)$, the potential outcomes $Y(w)$ are conditionally independent of treatment assignment given the observed features. This satisfies the unconfoundedness assumption by design.* **Bound-edness:** *The potential outcome $Y_i(W_i)$ represents binary classification correctness, thus naturally satisfying $0 \leq Y_i(W_i) \leq 1$ for all observations.*

**Remark 2.** *Beyond the structured-data applications examined in the main text, our framework also extends to unstructured domains such as NLP, computer vision, and recommender systems. Given their complexity and the primarily theoretical focus of our work, we provide only a high-level discussion in Appendix D, leaving detailed empirical studies for future work.*

## 5 DISCUSSION

We develop a hypothesis-testing framework for approximate fairness under explicit utility trade-offs. Our fairness criterion generalizes strong demographic parity, while expected utilities are defined using the potential-outcomes framework common in causal inference. To the best of our knowledge, this is the first theoretical framework to conduct hypothesis testing for fairness–utility trade-offs using a Wasserstein-projection approach. For future work, it would be interesting to explore Pareto-optimal frontiers of thresholds $(\epsilon, r)$.

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

## A   THE USE OF LARGE LANGUAGE MODELS (LLMs)

We occasionally used an LLM to refine the writing in the Introduction and some of the main context, but the main ideas and structure were entirely developed by the authors; the LLM served only as a tool to improve language flow.

## B   PROOFS

### B.1   PROOF OF STRONG DUALITY

In this section, we provide the proof for the first main result of the paper — Theorem 3.1.

*Proof of Theorem 3.1.* The Lagrangian function can be written as

$$
\begin{aligned}
&L(\lambda, \alpha; \nu) \\
&= \lambda r - \alpha \epsilon + \mathbb{E}_\nu \{c(X, X')\} \\
&\quad - \lambda \sum_{a \in \{0,1\}} p_a \mathbb{E}_\nu [m_1(X, a)\pi_a(X) + m_0(X, a)(1 - \pi_a(X))] \} \\
&\quad + \alpha \int_0^1 |\mathbb{E}_\nu \left[ \mathbf{1}\{\pi_1(X) \geq \tau\} - \mathbf{1}\{\pi_0(X) \geq \tau\} \right] | d\tau
\end{aligned}
\tag{12}
$$

where $\lambda \in \mathbb{R}_+$, $\alpha \in \mathbb{R}_+$, and $\nu$ belongs to the feasible set that

$$
\Gamma(\hat{\mathbb{P}}_N) = \left\{ \nu \in \mathcal{P}(\mathcal{X} \times \mathcal{X}) : \nu_{X'} = \hat{\mathbb{P}}_N \right\}.
$$

Note that $\mathcal{X}$ is compact, so $\mathcal{P}(\mathcal{X})$ is tight, so $\Gamma(\hat{\mathbb{P}}_N)$ is also tight. Note that $L(\lambda, \alpha; \nu)$ is convex in $\nu$ and linear in $(\lambda, \alpha)$. Thus $L(\lambda, \alpha; \nu)$ is a concave-cone mapping, where $L(\cdot; \nu)$ is concave and $L(\lambda, \alpha; \cdot)$ is convex.

We want to prove the following two statements:

1) The suprema of $\inf_{\nu \in \Gamma(\hat{\mathbb{P}}_N)} L(\lambda, \alpha; \nu)$ with respect to $(\lambda, \alpha)$ are bounded on $\mathbb{R}_+ \times \mathbb{R}_+$.

2) $L(\lambda, \alpha; \cdot)$ is lower bounded for some $(\lambda, \alpha)$ in the relative interior of some bounded subset of $\mathbb{R}_+ \times \mathbb{R}_+$.

To prove the first statement, let $\mathbb{Q}_0$ be a measure in $\mathcal{P}(\mathcal{X})$ such that $\mathbb{Q}_0$ concentrates on some $x \in \mathcal{X}$ (i.e. $\mathbb{Q}_0(X = x) = 1$), where $\pi_1(x) = \pi_0(x) = \xi \in (0, 1)$ and

$$
\sum_{a \in \{0,1\}} p_a(x)[m_1(x, a)\pi_a(x) + m_0(x, a)(1 - \pi_a(x))] \geq r.
$$

Then by taking $\nu_0 = \mathbb{Q}_0 \times \hat{\mathbb{P}}_N \in \Gamma(\hat{\mathbb{P}}_N)$, we have

$$
\sup_{(\lambda, \alpha) \in \mathbb{R}_+ \times \mathbb{R}_+} \inf_{\nu \in \Gamma(\hat{\mathbb{P}}_N)} L(\lambda, \alpha; \nu) \leq \sup_{(\lambda, \alpha) \in \mathbb{R}_+ \times \mathbb{R}_+} L(\lambda, \alpha; \nu_0),
$$

where

$$
\begin{aligned}
&\sup_{(\lambda, \alpha) \in \mathbb{R}_+ \times \mathbb{R}_+} L(\lambda, \alpha; \nu_0) \\
&= \mathbb{E}_{\nu_0}[c(X, X')] - \alpha \epsilon \\
&\quad + \lambda \{r - \sum_{a \in \{0,1\}} p_a(x)[m_1(x, a)\pi_a(x) + m_0(x, a)(1 - \pi_a(x))]\} \\
&= \mathbb{E}_{\nu_0}[c(X, X')],
\end{aligned}
\tag{13}
$$

where $\lambda^* = \alpha^* = 0$ in (13). Since $\mathcal{X}$ is compact and $c$ is continuous, thus $\mathbb{E}_{\nu_0}[c(X, X')]$ is bounded. Hence

$$
\sup_{(\lambda, \alpha) \in \mathbb{R}_+ \times \mathbb{R}_+} \inf_{\nu \in \Gamma(\hat{\mathbb{P}}_N)} L(\lambda, \alpha; \nu) < \infty.
$$

Assume that the suprema of $\inf_{\nu \in \Gamma(\hat{\mathbb{P}}_N)} L(\lambda, \alpha; \nu)$ with respect to $\lambda, \alpha$ goes to infinity in

$$
\sup_{(\lambda, \alpha) \in \mathbb{R}_+ \times \mathbb{R}_+} \inf_{\nu \in \Gamma(\hat{\mathbb{P}}_N)} L(\lambda, \alpha; \nu),
$$

since for any $\nu \in \Gamma(\hat{\mathbb{P}}_N)$,

$$
\begin{aligned}
&L(\lambda, \alpha; \nu) \\
&= \mathbb{E}_\nu[c(X, X')] + \lambda r + \alpha \left\{ \int_0^1 |\mathbb{E}_\nu[\mathbf{1}\{\pi_1(X) \geq \tau\} - \mathbf{1}\{\pi_0(X) \geq \tau\}]| - \epsilon \right\} \\
&\quad - \lambda \sum_{a \in \{0,1\}} \mathbb{E}_\nu[\{m_1(X, a)\pi_a(X) + m_0(X, a)(1 - \pi_a(X))\}p_a(X)]
\end{aligned}
\tag{14}
$$

and we already know that

$$
\sup_{(\lambda, \alpha) \in \mathbb{R}_+ \times \mathbb{R}_+} \inf_{\nu \in \Gamma(\hat{\mathbb{P}}_N)} L(\lambda, \alpha; \nu) < \infty,
$$

thus given any

$$
(\lambda_j, \alpha_j) \in \mathbb{R}_+ \times \mathbb{R}_+,
$$

such that either $\lambda_j \to \infty$ or $\alpha_j \to \infty$ holds as $j \to \infty$, let

$$
\{\nu_k^j\}_{k \in \mathbb{N}} \subset \Gamma(\hat{\mathbb{P}}_N)
$$

be a sequence of probability measures such that

$$
\begin{aligned}
&\lim_{j \to \infty} \lim_{k \to \infty} L(\lambda_j, \alpha_j; \nu_k^j) \\
&= \lim_{j \to \infty} \inf_{\nu \in \Gamma(\hat{\mathbb{P}}_N)} L(\lambda_j, \alpha_j; \nu) \\
&= \sup_{(\lambda, \alpha) \in \mathbb{R}_+ \times \mathbb{R}_+} \inf_{\nu \in \Gamma(\hat{\mathbb{P}}_N)} L(\lambda, \alpha; \nu) < \infty.
\end{aligned}
$$

Thus there must exist some $J$, such that for any $j \geq J$ and for any $k \in \mathbb{N}$, we have

$$
\int_0^1 |\mathbb{E}_{\nu_k^j}[\mathbf{1}\{\pi_1(X) \geq \tau\} - \mathbf{1}\{\pi_0(X) \geq \tau\}]| - \epsilon \leq 0.
$$

$$
r - \sum_{a \in \{0,1\}} \mathbb{E}_{\nu_k^j}[\{m_1(X, a)\pi_a(X) + m_0(X, a)(1 - \pi_a(X))\}p_a(X)] \leq 0.
$$

Suppose there exists subsequences $\{j_n\} \subset \mathbb{N}$ where $j_n \geq J$ there are infinitely many $k$ such that at least one of the following two strict inequalities hold:

$$
\int_0^1 |\mathbb{E}_{\nu_k^{j_n}}[\mathbf{1}\{\pi_1(X) \geq \tau\} - \mathbf{1}\{\pi_0(X) \geq \tau\}]| - \epsilon < 0,
$$

$$
r - \sum_{a \in \{0,1\}} p_a \mathbb{E}_{\nu_k^{j_n}}[m_1(X, a)\pi_a(X) + m_0(X, a)(1 - \pi_a(X))] < 0.
$$

Note that $\lambda_{j_n}, \alpha_{j_n} \to \infty$, then we have a subsequence $\{\lambda_{j_n}\} \subset \{\lambda_j\}$, $\{\alpha_{j_n}\} \subset \{\alpha_j\}$, such that

$$
\begin{aligned}
-\infty \quad &= \lim_{j_n \to \infty} \inf_{\nu \in \Gamma(\hat{\mathbb{P}}_N)} L(\lambda_{j_n}, \alpha_{j_n}; \nu) \\
&= \sup_{(\lambda, \alpha) \in \mathbb{R}_+ \times \mathbb{R}_+} \inf_{\nu \in \Gamma(\hat{\mathbb{P}}_N)} L(\lambda, \alpha; \nu) \\
&\geq L(0, 0; \nu) > -\infty,
\end{aligned}
$$

which leads to contradiction. Hence for any $j$, we can only have finitely many $k$ for where one of the following strict inequality holds:

$$
\int_0^1 |\mathbb{E}_{\nu_k^j}[\mathbf{1}\{\pi_1(X) \geq \tau\} - \mathbf{1}\{\pi_0(X) \geq \tau\}]| - \epsilon < 0,
$$

$$
r - \sum_{a \in \{0,1\}} \mathbb{E}_{\nu_k^j}[\{m_1(X, a)\pi_a(X) + m_0(X, a)(1 - \pi_a(X))\}p_a(X)] < 0.
$$

This implies that for any j, except for at most finitely many $k$, we have

$$
\int_0^1 |\mathbb{E}_{\nu_k^j}[\mathbf{1}\{\pi_1(X) \geq \tau\} - \mathbf{1}\{\pi_0(X) \geq \tau\}]| - \epsilon = 0,
$$

$$
r - \sum_{a \in \{0,1\}} \mathbb{E}_{\nu_k^j}[\{m_1(X, a)\pi_a(X) + m_0(X, a)(1 - \pi_a(X))\}p_a(X)] = 0.
$$

This implies that we can take $\Lambda \subset \mathbb{R}_+, \mathcal{S} \subset \mathbb{R}_+$, where $\Lambda = [0, B], \mathcal{S} = [0, B]$, and $B$ is a sufficiently large but bounded constant, we have

$$\sup_{(\lambda,\alpha)\in\mathbb{R}_+\times\mathbb{R}_+} \inf_{\nu\in\Gamma(\hat{\mathbb{P}}_N)} L(\lambda,\alpha;\nu) = \sup_{(\lambda,\alpha)\in\Lambda\times\mathcal{S}} \inf_{\nu\in\Gamma(\hat{\mathbb{P}}_N)} L(\lambda,\alpha;\nu). \tag{15}$$

Thus we have proved the first statement.

To prove the second statement, it is sufficient to prove that given some $\lambda > 0, \alpha > 0, L(\lambda,\alpha;\nu)$ is lower bounded for any $\nu \in \Gamma(\hat{\mathbb{P}}_N)$. This follows immediately by (14), the compactness of $\mathcal{X}$ and the continuity of $c, \pi_1, \pi_0, m_1(\cdot,1), m_0(\cdot,0)$. Thus by Lemma B.2, we have

$$\begin{aligned} &\sup_{(\lambda,\alpha)\in\mathbb{R}_+\times\mathbb{R}_+} \inf_{\nu\in\Gamma(\hat{\mathbb{P}}_N)} L(\lambda,\alpha;\nu) \\ &= \sup_{(\lambda,\alpha)\in\Lambda\times\mathcal{S}} \inf_{\nu\in\Gamma(\hat{\mathbb{P}}_N)} L(\lambda,\alpha;\nu) \\ &= \inf_{\nu\in\Gamma(\hat{\mathbb{P}}_N)} \sup_{(\lambda,\alpha)\in\Lambda\times\mathcal{S}} L(\lambda,\alpha;\nu). \end{aligned} \tag{16}$$

For the last step, we want to show that for $B$ large enough, with $\Lambda = [0, B], \mathcal{S} = [0, B]$, we have

$$\begin{aligned} &\inf_{\nu\in\Gamma(\hat{\mathbb{P}}_N)} \sup_{(\lambda,\alpha)\in\Lambda\times\mathcal{S}} L(\lambda,\alpha;\nu) \\ &= \inf_{\nu\in\Gamma(\hat{\mathbb{P}}_N)} \sup_{(\lambda,\alpha)\in\mathbb{R}_+\times\mathbb{R}_+} L(\lambda,\alpha;\nu). \end{aligned} \tag{17}$$

First note that when $\alpha \to \infty$ or $\lambda \to \infty$, by taking $\nu_0 = \mathbb{Q}_0 \times \hat{\mathbb{P}}_N$, where $\mathbb{Q}_0$ is defined in the same way as before, we will have

(i) $\displaystyle\inf_{\nu\in\Gamma(\hat{\mathbb{P}}_N)} \lim_{\lambda\to\infty,\alpha\to\infty} L(\lambda,\alpha;\nu) \le \lim_{\lambda\to\infty,\alpha\to\infty} L(\lambda,\alpha;\nu_0) = -\infty.$

(ii) $\displaystyle\inf_{\nu\in\Gamma(\hat{\mathbb{P}}_N)} \lim_{\lambda\to\infty} L(\lambda,\alpha;\nu) \le \lim_{\lambda\to\infty} L(\lambda,\alpha;\nu_0) = -\infty$ fixing any $\alpha \ge 0$

(iii) $\displaystyle\inf_{\nu\in\Gamma(\hat{\mathbb{P}}_N)} \lim_{\alpha\to\infty} L(\lambda,\alpha;\nu) \le \lim_{\alpha\to\infty} L(\lambda,\alpha;\nu_0) = -\infty$ fixing any $\lambda \ge 0$.

And note that

$$\begin{aligned} &\inf_{\nu\in\Gamma(\hat{\mathbb{P}}_N)} \sup_{(\lambda,\alpha)\in\Lambda\times\mathcal{S}} L(\lambda,\alpha,\nu) \\ &\ge \inf_{\nu\in\Gamma(\hat{\mathbb{P}}_N)} L(0,0,\nu) \\ &= \inf_{\nu\in\Gamma(\hat{\mathbb{P}}_N)} \mathbb{E}_\nu[c(X,X')] > -\infty. \end{aligned} \tag{18}$$

Suppose (17) does not hold for any $B > 0$. Then for any $B > 0$, for any $(\lambda,\alpha) \in [0, B] \times [0, B]$, there always exists some $\lambda_1 > B$ or $\alpha_1 > B$, such that at least one the three statements holds:

(a) $\displaystyle\inf_{\nu\in\Gamma(\hat{\mathbb{P}}_N)} \sup_{(\lambda,\alpha)\in\Lambda\times\mathcal{S}} L(\lambda,\alpha;\nu) < \inf_{\nu\in\Gamma(\hat{\mathbb{P}}_N)} L(\lambda_1,\alpha_1;\nu);$

(b) $\displaystyle\inf_{\nu\in\Gamma(\hat{\mathbb{P}}_N)} \sup_{(\lambda,\alpha)\in\Lambda\times\mathcal{S}} L(\lambda,\alpha;\nu) < \inf_{\nu\in\Gamma(\hat{\mathbb{P}}_N)} L(\lambda_1,\alpha;\nu)$ fixing any $\alpha \ge 0$;

(c) $\displaystyle\inf_{\nu\in\Gamma(\hat{\mathbb{P}}_N)} \sup_{(\lambda,\alpha)\in\Lambda\times\mathcal{S}} L(\lambda,\alpha;\nu) < \inf_{\nu\in\Gamma(\hat{\mathbb{P}}_N)} L(\lambda,\alpha_1;\nu)$ fixing any $\lambda \ge 0$;

By letting $M \to \infty$ and inequality (18), we can see that statement (a) violates statement (i), (b) violates (ii) and (c) violates (iii). Hence (17) holds for some $B > 0$ sufficiently large. Then together with (16), we have

$$\sup_{(\lambda,\alpha)\in\mathbb{R}_+\times\mathbb{R}_+} \inf_{\nu\in\Gamma(\hat{\mathbb{P}}_N)} L(\lambda,\alpha;\nu) = \inf_{\nu\in\Gamma(\hat{\mathbb{P}}_N)} \sup_{(\lambda,\alpha)\in\mathbb{R}_+\times\mathbb{R}_+} L(\lambda,\alpha;\nu).$$

As a result we have

$$
\mathcal{R}_{r,\epsilon}(\hat{\mathbb{P}}_N)
$$
$$
= \sup_{(\lambda,\alpha)\in\mathbb{R}_+\times\mathbb{R}_+} \inf_{\nu\in\Gamma(\hat{\mathbb{P}}_N)} \mathbb{E}_\nu[c(X,X')] + \lambda r
$$
$$
+ \alpha\left\{ \int_0^1 |\mathbb{E}_\nu[\mathbf{1}\{\pi_1(X)\geq\tau\} - \mathbf{1}\{\pi_0(X)\geq\tau\}]|d\tau - \epsilon \right\}
$$
$$
- \lambda \sum_{a\in\{0,1\}} \mathbb{E}_\nu[\{m_1(X,a)\pi_a(X) + m_0(X,a)(1-\pi_a(X))\}p_a(X)]\}
$$
$$
=_{(a)} \sup_{(\lambda,\alpha)\in\mathbb{R}_+\times\mathbb{R}_+} \inf_{\nu\in\Gamma(\hat{\mathbb{P}}_N)} \mathbb{E}_\nu[c(X,X')] + \alpha\{\mathbb{E}_\nu[|\pi_1(X)-\pi_0(X)|] - \epsilon\}
$$
$$
+ \lambda\left\{ r - \sum_{a\in\{0,1\}} \mathbb{E}_\nu[\{m_1(X,a)\pi_a(X) + m_0(X,a)(1-\pi_a(X))\}p_a(X)] \right\}
$$
$$
=_{(b)} \sup_{(\lambda,\alpha)\in\mathbb{R}_+\times\mathbb{R}_+} \lambda r - \alpha\epsilon + \frac{1}{N}\sum_{i=1}^N \min_{x\in\mathcal{X}}\{\|x-X_i\|^2 + \alpha|\pi_1(x)-\pi_0(x)| - \lambda M(x)\}.
$$

where (a) follows from Lemma B.3, and in (b)

$$
M(x) = \sum_{a\in\{0,1\}} p_a(x)[m_1(x,a)\pi_a(x) + m_0(x,a)(1-\pi_a(x))].
$$

□

## B.2 USEFUL LEMMAS

**Lemma B.1** (Proposition 1 of Jiang et al. (2020))**.** *Let*

$$
\mathcal{J} = \left\{ J:[0,1]\to[0,1] \middle| \int_{\mathcal{B}} p_{\pi_1(X_i)}(y)dy = \int_{J^{-1}(\mathcal{B})} p_{\pi_0(X_i)}(x)dx, \ \forall \text{ measurable } \mathcal{B}\subset[0,1] \right\}.
$$

*The following two quantities are equal:*

*(i)* $\mathcal{W}_1(p_{\pi_1(X_i)}, p_{\pi_0(X_i)}) = \min_{J\in\mathcal{J}} \int_{x\in[0,1]} |x-J(x)|p_{S_{\pi_0}(X_i)}(x)dx.$

*(ii)* $\mathbb{E}_{\tau\sim\text{Unif}[0,1]}|\mathbb{P}(\pi_1(X_i)>\tau) - \mathbb{P}(\pi_0(X_i)>\tau)|.$

The proof of Lemma B.1 follows directly from Proposition 1 of Jiang et al. (2020).

**Lemma B.2** (Theorem 1 of Vianney & Vigeral (2015))**.** *Let $\mathcal{Z}_1$ and $\mathcal{Z}_2$ be two nonempty convex sets and $f : \mathcal{Z}_1\times\mathcal{Z}_2 \to \mathbb{R}$ be a concave-convex mapping, i.e. $f(\cdot,z_2)$ is concave and $f(z_1,\cdot)$ is convex for every $z_1\in\mathcal{Z}_1$ and $z_2\in\mathcal{Z}_2$. Assume that*

- $\mathcal{Z}_1$ *is finite-dimensional.*

- $\mathcal{Z}_2$ *is bounded.*

- $f(z_1,\cdot)$ *is lower bounded for some $z_1$ in the relative interior of $\mathcal{Z}_1$.*

*Then*
$$
\sup_{z\in\mathcal{Z}_1} \inf_{z_2\in\mathcal{Z}_2} f(z_1,z_2) = \inf_{z_2\in\mathcal{Z}_2} \sup_{z_1\in\mathcal{Z}_1} f(z_1,z_2).
$$

**Lemma B.3.** *Under Assumptions 2, 3, for any $\nu\in\Gamma(\hat{\mathbb{P}}_N)$, we have*

$$
\int_0^1 |\nu(\pi_1(X)>\tau) - \nu(\pi_0(X)>\tau)|d\tau = \mathbb{E}_\nu[|\pi_1(X)-\pi_0(X)|].
$$

*Proof of Lemma B.3.* For $X \sim \mathbb{Q}$, let $\nu_1$ be the distribution of $\pi_1(X)$ and $\nu_0$ be the distribution of $\pi_0(X)$. Then

$$
\begin{aligned}
\mathcal{V} &:= \int_0^1 |\mathbb{Q}(\pi_1(X) > \tau) - \mathbb{Q}(\pi_0(X) > \tau)|d\tau = \mathcal{W}_1(\nu_1, \nu_0) \\
&= \inf_{\pi \in \Pi(\nu_1, \nu_0)} \mathbb{E}_\pi[|Z - Z'|],
\end{aligned}
\tag{19}
$$

where $\nu_1, \nu_0 \in \mathcal{P}([0,1])$, and $\mathcal{W}_1$ is the 1-Wasserstein distance. Denote

$$
\mathcal{S} = \{(\alpha, \beta) | (\alpha, \beta) \in \mathcal{C}([0,1]) \times \mathcal{C}([0,1]) : \alpha(z) + \beta(z') \leq |z - z'|\},
$$

where $\mathcal{C}([0,1])$ is the collection of continuous functions on $[0,1]$. The dual formulation to the Kantorovich's problem of (19) can be written as

$$
\begin{aligned}
\mathcal{D} &= \sup_{(\alpha,\beta) \in \mathcal{S}} \mathbb{E}_{\nu_1}[\alpha(Z)] + \mathbb{E}_{\nu_0}[\beta(Z')] \\
&=_{(1)} \sup_{(\alpha,\beta) \in \mathcal{S}} \mathbb{E}_\mathbb{Q}[\alpha(\pi_1(X)) + \beta(\pi_0(X))] \\
&=_{(2)} \mathbb{E}_\mathbb{Q}[|\pi_1(X) - \pi_0(X)|],
\end{aligned}
$$

where (1) follows because

$$
\mathbb{E}_{\nu_1}[\alpha(Z)] = \mathbb{E}_\mathbb{Q}[\alpha(\pi_1(X))], \quad \mathbb{E}_{\nu_0}[\beta(Z')] = \mathbb{E}_\mathbb{Q}[\beta(\pi_0(X))],
$$

and (2) follows since the optimal $\alpha(\cdot)$, $\beta(\cdot)$ satisfy

$$
\alpha^*(z) + \beta^*(z') = |z - z'|
$$

for almost surely $(z, z') \in [0,1] \times [0,1]$. By strong duality Villani et al. (2009), we have $\mathcal{V} = \mathcal{D}$, where $\mathcal{V}$ is defined in (19). So

$$
\int_0^1 |\mathbb{Q}(\pi_1(X) > \tau) - \mathbb{Q}(\pi_0(X) > \tau)|d\tau = \mathbb{E}_\mathbb{Q}[|\pi_1(X) - \pi_0(X)|].
\tag{20}
$$

Note that for any $\nu \in \Gamma(\hat{\mathbb{P}}_N)$ with $\nu_{X'} = \hat{\mathbb{P}}_N$, we have

$$
\begin{aligned}
&\int_0^1 |\nu(\pi_1(X) > \tau) - \nu(\pi_0(X) > \tau)|d\tau \\
&= \int_0^1 |\nu_X(\pi_1(X) > \tau) - \nu_X(\pi_0(X) > \tau)|d\tau,
\end{aligned}
$$

and

$$
\mathbb{E}_\nu[|\pi_1(X) - \pi_0(X)|] = \mathbb{E}_{\nu_X}[|\pi_1(X) - \pi_0(X)|].
$$

Note that (20) holds for arbitrary $\mathbb{Q} \in \mathcal{P}(\mathcal{X})$, thus the result follows. $\qquad\square$

### B.3 PROOF OF THEOREM 3.2

Recall from Theorem 3.1 that

$$
\mathcal{R}_{r,\epsilon}(\hat{\mathbb{P}}_N) = \sup_{(\lambda, \alpha) \in \mathbb{R}_+ \times \mathbb{R}_+} \lambda r - \alpha\epsilon
$$

$$
+ \frac{1}{N} \sum_{i=1}^N \min_{x \in \mathcal{X}} \{\|x - X_i\|^2 + \alpha|\pi_1(x) - \pi_0(x)| - \lambda M(x)\},
$$

where $M(x) = \sum_{a \in \{0,1\}} p_a(x)[m_1(x, a)\pi_a(x) + m_0(x, a)(1 - \pi_a(x))]$ and $c(x, y) = \|x - y\|$.

Change variables as $\Delta = x - X_i$, by fundamental theorem of calculus and Assumption 2, we have

$$
\begin{aligned}
\pi_1(x) - \pi_1(X_i) &= \int_0^1 D\pi_1(X_i + u\Delta)\Delta du, \\
\pi_0(x) - \pi_0(X_i) &= \int_0^1 D\pi_0(X_i + u\Delta)\Delta du,
\end{aligned}
$$

thus

$$|\pi_1(x) - \pi_0(x)| = \left| \int_0^1 [D\pi_1(X_i + u\Delta) - D\pi_0(X_i + u\Delta)]\Delta du + (\pi_1(X_i) - \pi_0(X_i)) \right|.$$

Additionally,

$$M(X_i + \Delta) - M(X_i) = \int_0^1 DM(X_i + u\Delta)\Delta du.$$

So

$$\mathcal{R}_{r,\epsilon}(\hat{\mathbb{P}}_N)$$

$$= \sup_{(\bar{\lambda},\bar{\alpha})\in\mathbb{R}_+\times\mathbb{R}_+} \bar{\lambda}r - \bar{\alpha}\epsilon - \bar{\lambda}\cdot\frac{1}{N}\sum_{i=1}^N M(X_i)$$

$$+ \frac{1}{N}\sum_{i=1}^N \min_\Delta \left\{ \|\Delta\| + \bar{\alpha}\left| \int_0^1 [D(\pi_1 - \pi_0)(X_i + u\Delta)]\Delta du + (\pi_1(X_i) - \pi_0(X_i)) \right| \right.$$

$$\left. - \bar{\lambda}\int_0^1 DM(X_i + u\Delta)\Delta du \right\}$$

$$= \sup_{(\bar{\lambda},\bar{\alpha})\in\mathbb{R}_+\times\mathbb{R}_+} \lambda\cdot\frac{1}{N}\sum_{i=1}^N \{(r - M(X_i)) - \mathbb{E}[r - M(X_i)]\} - \bar{\alpha}\epsilon + \bar{\lambda}\mathbb{E}[r - M(X_i)]$$

$$+ \frac{1}{N}\sum_{i=1}^N \min_\Delta \left\{ \|\Delta\|^2 + \bar{\alpha}\left| \int_0^1 [D(\pi_1 - \pi_0)(X_i + u\Delta)]\Delta du + (\pi_1(X_i) - \pi_0(X_i)) \right| \right.$$

$$\left. - \bar{\lambda}\int_0^1 DM(X_i + u\Delta)\Delta du \right\}.$$

Then redefining $\Delta = \Delta/N^{1/2}$, $\lambda = \sqrt{N}\bar{\lambda}$, $\alpha = \sqrt{N}\bar{\alpha}$, we have

$$N\mathcal{R}_{r,\epsilon}(\hat{\mathbb{P}}_N) \quad = \sup_{(\lambda,\alpha)\in\mathbb{R}_+\times\mathbb{R}_+} \lambda M_N(r) + \mathcal{E}_N(\alpha, \lambda) \tag{21}$$
$$+ \lambda\sqrt{N}\mathbb{E}[r - M(X_i)] - \alpha\sqrt{N}\epsilon,$$

where

$$\mathcal{E}_N(\alpha, \lambda)$$

$$= \frac{1}{N}\sum_{i=1}^N \min_\Delta \left\{ \|\Delta\|^2 - \lambda\int_0^1 DM(X_i + N^{-1/2}\Delta u)\Delta du \right. \tag{22}$$

$$\left. + \alpha\left| \int_0^1 [D(\pi_1 - \pi_0)(X_i + N^{-1/2}\Delta u)]\Delta du + \sqrt{N}(\pi_1(X_i) - \pi_0(X_i)) \right| \right\},$$

and

$$M_N(r) = \frac{1}{\sqrt{N}}\sum_{i=1}^N \{(r - M(X_i)) - \mathbb{E}[r - M(X_i)]\}.$$

Denote

$$\bar{R}(\alpha, \lambda) = \lambda M_N(r) + \mathcal{E}_N(\alpha, \lambda) + \lambda\sqrt{N}\mathbb{E}[r - M(X_i)] - \alpha\sqrt{N}\epsilon.$$

Note that the right hand side of (21) is non-negative, because

$$\sup_{(\lambda,\alpha)\in\mathbb{R}_+\times\mathbb{R}_+} \bar{R}(\alpha, \lambda) \geq \bar{R}(0, 0) \geq 0.$$

By (15) in the proof of Theorem 3.1, For $\Lambda = [0, B], \mathcal{S} = [0, B]$ where $B$ is a sufficiently large constant, we have

$$\sup_{(\lambda,\alpha)\in\mathbb{R}_+\times\mathbb{R}_+} \inf_{\nu\in\Gamma(\hat{\mathbb{P}}_N)} L(\lambda, \alpha; \nu) = \sup_{(\lambda,\alpha)\in\Lambda\times\mathcal{S}} \inf_{\nu\in\Gamma(\hat{\mathbb{P}}_N)} L(\lambda, \alpha; \nu). \tag{23}$$

So we can constrain the optimization with respect of $(\lambda, \alpha) \in \mathbb{R}_+ \times \mathbb{R}_+$ within $\Lambda \times \mathcal{S}$.

For the summands in (22), we have

$$\min_\Delta \left\{ \|\Delta\|^2 + \alpha \left| \int_0^1 \left[ D\pi_1\left(X_i + \frac{\Delta u}{\sqrt{N}}\right) - D\pi_0\left(X_i + \frac{\Delta u}{\sqrt{N}}\right) \right] \Delta du \right. \right.$$
$$\left. + \sqrt{N}(\pi_1(X_i) - \pi_0(X_i)) \right|$$
$$\left. - \lambda \int_0^1 DM(X_i + N^{-1/2}\Delta u) \Delta du \right\}$$
$$= \min_\Delta \left\{ \|\Delta\|^2 + \alpha \left| \int_0^1 [D\pi_1(X_i + N^{-1/2}\Delta u) - D\pi_1(X_i)] \Delta du \right. \right.$$
$$- \int_0^1 [D\pi_0(X_i + N^{-1/2}\Delta u) - D\pi_0(X_i)] \Delta du$$
$$\left. + \sqrt{N}(\pi_1(X_i) - \pi_0(X_i)) + [D(\pi_1 - \pi_0)(X_i)]\Delta \right|$$
$$- \lambda \int_0^1 [DM(X_i + N^{-1/2}\Delta u) - DM(X_i)] \Delta du$$
$$\left. - \lambda DM(X_i)\Delta \right\}$$
$$=_{(a)} \min_\Delta \left\{ \|\Delta\|^2 + \alpha|[D(\pi_1 - \pi_0)(X_i)]\Delta + \sqrt{N}(\pi_1(X_i) - \pi_0(X_i))| - \lambda DM(X_i)\Delta + R_i \right\}$$
$$(24)$$

where

$$R_i = \alpha \left| \int_0^1 [D\pi_1(X_i + N^{-1/2}\Delta u) - D\pi_1(X_i)] \Delta du \right|$$
$$+ \alpha \left| \int_0^1 [D\pi_0(X_i + N^{-1/2}\Delta u) - D\pi_0(X_i)] \Delta du \right|$$
$$+ \lambda \left| \int_0^1 [DM(X_i + N^{-1/2}\Delta u) - DM(X_i)] \Delta du \right|.$$

By Assumption 2 and the continuity of $D\pi_1(\cdot)$, $D\pi_0(\cdot)$, $DM(\cdot)$, we have

$$\frac{1}{N} \sum_{i=1}^N R_i \Rightarrow 0 \qquad (25)$$

uniformly over $\Delta$ in a compact set, $\lambda \in [0, B]$ and $\alpha \in [0, B]$, as $n \to \infty$. Thus by (21),

$$N\mathcal{R}_{r,\epsilon}(\hat{\mathbb{P}}_N)$$
$$= \sup_{(\lambda,\alpha) \in \mathbb{R}_+ \times \mathbb{R}_+} \lambda M_N(r) + \lambda\sqrt{N}\{r - \mathbb{E}[M(X_i)]\} - \alpha\sqrt{N}\epsilon$$
$$+ \frac{1}{N} \sum_{i=1}^N \min_\Delta \left\{ \|\Delta\|^2 + \alpha|[D(\pi_1 - \pi_0)(X_i)]\Delta + \sqrt{N}(\pi_1(X_i) - \pi_0(X_i))| \right.$$
$$\left. - \lambda DM(X_i)\Delta + R_i \right\}$$
$$\leq \sup_{(\lambda,\alpha) \in \mathbb{R}_+ \times \mathbb{R}_+} \lambda M_N(r) + \alpha\Pi_N(\epsilon) + \lambda\sqrt{N}\{r - \mathbb{E}[M(X_i)]\} + \alpha\sqrt{N}\{\mathbb{E}[|\pi_1(X_i) - \pi_0(X_i)|] - \epsilon\}$$
$$+ \frac{1}{N} \sum_{i=1}^N \min_\Delta \left\{ \|\Delta\|^2 - \lambda DM(X_i)\Delta + R_i \right.$$
$$\left. + \alpha \cdot \mathrm{sgn}\left([D(\pi_1 - \pi_0)(X_i)]\Delta\right)[D(\pi_1 - \pi_0)(X_i)]\Delta \right\},$$
$$(26)$$

where

$$\Pi_N(\epsilon) = \frac{1}{N} \sum_{i=1}^N |\pi_1(X_i) - \pi_0(X_i)| - \mathbb{E}[|\pi_1(X_i) - \pi_0(X_i)|].$$

Note that if $[D(\pi_1 - \pi_0)(X_i)]\Delta \geq 0$, then

$$\|\Delta\|^2 + \alpha \cdot \mathrm{sgn}\left([D(\pi_1 - \pi_0)(X_i)]\Delta\right)[D(\pi_1 - \pi_0)(X_i)]\Delta - \lambda DM(X_i)\Delta$$
$$= \|\Delta\|^2 + [\alpha\{D(\pi_1 - \pi_0)(X_i)\} - \lambda DM(X_i)]\Delta.$$

If $[D(\pi_1 - \pi_0)(X_i)]\Delta < 0$, then

$$\|\Delta\|^2 + \alpha \cdot \text{sgn}\left([D(\pi_1 - \pi_0)(X_i)]\Delta\right)[D(\pi_1 - \pi_0)(X_i)]\Delta - \lambda DM(X_i)\Delta$$
$$= \|\Delta\|^2 - [\alpha\{D(\pi_1 - \pi_0)(X_i)\} + \lambda DM(X_i)]\Delta$$

Note that

$$\arg\min_{\Delta} \|\Delta\|^2 + [\alpha\{D(\pi_1 - \pi_0)(X_i)\} - \lambda DM(X_i)]\Delta$$
$$= \frac{\lambda DM(X_i) - \alpha D[\pi_1(X_i) - \pi_0(X_i)]}{2},$$

$$\arg\min_{\Delta} \|\Delta\|^2 - [\alpha\{D(\pi_1 - \pi_0)(X_i)\} + \lambda DM(X_i)]\Delta$$
$$= \frac{\lambda DM(X_i) + \alpha D[\pi_1(X_i) - \pi_0(X_i)]}{2}.$$

So we have

$$\min_{\Delta} \Big\{ \|\Delta\|^2$$
$$+ \alpha \cdot \text{sgn}\left([D(\pi_1 - \pi_0)(X_i)]\Delta\right)[D(\pi_1 - \pi_0)(X_i)]\Delta$$
$$- \lambda DM(X_i)\Delta \Big\}$$
$$\leq \min \left\{ \begin{array}{c} -1/4\|\lambda DM(X_i) - \alpha[D(\pi_1 - \pi_0)(X_i)]\|^2 \mathbf{1}_{\mathcal{E}_+}, \\ -1/4\|\lambda DM(X_i) + \alpha[D(\pi_1 - \pi_0)(X_i)]\|^2 \mathbf{1}_{\mathcal{E}_-} \end{array} \right\}$$

where $\mathcal{E}^+$ and $\mathcal{E}^-$ denote the events

$$\mathcal{E}^+ = \left\{ \begin{array}{c} \lambda DM(X_i)'[D(\pi_1 - \pi_0)(X_i)] \\ \geq \alpha\|D(\pi_1 - \pi_0)(X_i)\|^2 \end{array} \right\},$$

$$\mathcal{E}^- = \left\{ \begin{array}{c} \lambda DM(X_i)'[D(\pi_1 - \pi_0)(X_i)] \\ < -\alpha\|D(\pi_1 - \pi_0)(X_i)\|^2 \end{array} \right\}.$$

So by (26), we have

$$N\mathcal{R}_{r,\epsilon}(\hat{\mathbb{P}}_N)$$
$$\leq \max_{(\lambda,\alpha)\in\Lambda\times\mathcal{S}} \lambda M_N(r) + \alpha\Pi_N(\epsilon) + \lambda\sqrt{N}\{r - \mathbb{E}[M(X_i)]\}$$
$$+ \alpha\sqrt{N}\{\mathbb{E}[|\pi_1(X_i) - \pi_0(X_i)|] - \epsilon\}$$
$$+ \frac{1}{N}\sum_{i=1}^N \min \Big\{ \left(-\frac{1}{4}\|\lambda DM(X_i) - \alpha[D(\pi_1 - \pi_0)(X_i)]\|^2 + R_i\right)\mathbf{1}_{\mathcal{E}_+},$$
$$\left(-\frac{1}{4}\|\lambda DM(X_i) + \alpha[D(\pi_1 - \pi_0)(X_i)]\|^2 + R_i\right)\mathbf{1}_{\mathcal{E}_-} \Big\}.$$

So let $r^* = \mathbb{E}[M(X_i)]$, $\epsilon^* = \mathbb{E}[|\pi_1(X_i) - \pi_0(X_i)|]$, according to (25) we have

$$\max_{(\lambda,\alpha)\in\Lambda\times\mathcal{S}} \lambda M_N(r) + \alpha\Pi_N + \sqrt{N}\{\lambda(r - r^*) + \alpha(\epsilon^* - \epsilon)\}$$
$$+ \frac{1}{N}\sum_{i=1}^N \min \Big\{ \left(-\frac{1}{4}\|\lambda DM(X_i) - \alpha[D(\pi_1 - \pi_0)(X_i)]\|^2 + R_i\right)\mathbf{1}_{\mathcal{E}_+},$$
$$\left(-\frac{1}{4}\|\lambda DM(X_i) + \alpha[D(\pi_1 - \pi_0)(X_i)]\|^2 + R_i\right)\mathbf{1}_{\mathcal{E}_-} \Big\}$$
$$\Rightarrow \sup_{(\lambda,\alpha)\in\mathbb{R}_+\times\mathbb{R}_+:\lambda(r-r^*)+\alpha(\epsilon^*-\epsilon)=0} \lambda\overline{M} + \alpha\overline{\Pi} + \mathbb{E}[\bar{Z}(\lambda,\alpha)],$$

where

$$\overline{M} \sim \mathcal{N}(0, \text{cov}[M(X_i)]), \quad \overline{\Pi} \sim \mathcal{N}(0, \text{cov}[|\pi_1(X_i) - \pi_0(X_i)|]),$$

and

$$\bar{Z}(\lambda,\alpha) = \min \left\{ \begin{array}{c} -1/4\|\lambda DM(X_i) - \alpha[D(\pi_1 - \pi_0)(X_i)]\|^2\mathbf{1}_{\mathcal{E}_+}, \\ -1/4\|\lambda DM(X_i) + \alpha[D(\pi_1 - \pi_0)(X_i)]\|^2\mathbf{1}_{\mathcal{E}_-} \end{array} \right\}$$

Hence by (26) we have

$$
\begin{aligned}
& N\mathcal{R}_{r,\epsilon}(\hat{\mathbb{P}}_N) \\
& \lesssim_D \sup_{(\lambda,\alpha)\in\mathbb{R}_+\times\mathbb{R}_+:\lambda(r-r^*)+\alpha(\epsilon^*-\epsilon)=0} \lambda\overline{M} + \alpha\overline{\Pi} + \mathbb{E}[\bar{Z}(\lambda,\alpha)].
\end{aligned}
$$

By Fatou's Lemma, letting $\zeta = (\lambda,\alpha)$,

$$
S_+ = \begin{pmatrix} DM(X_i) \\ -D[\pi_1 - \pi_0](X_i) \end{pmatrix}, \quad S_- = \begin{pmatrix} DM(X_i) \\ D[\pi_1 - \pi_0](X_i) \end{pmatrix},
$$

then we have

$$
\mathbb{E}[\bar{Z}(\lambda,\alpha)] \le \min\left\{ -\tfrac{1}{4}\zeta^T\mathbb{E}[S_+ S_+^T \mathbf{1}_{\mathcal{E}^+}]\zeta, -\tfrac{1}{4}\zeta^T\mathbb{E}[S_- S_-^T \mathbf{1}_{\mathcal{E}^-}]\zeta \right\}
$$

Let $\overline{\mathcal{W}} = \begin{pmatrix} \overline{M} \\ \overline{\Pi} \end{pmatrix}$, then we have

$$
N\mathcal{R}_{r,\epsilon}(\hat{\mathbb{P}}_N) \lesssim_D \sup_{\zeta\ge\mathbf{0}} \zeta^T\overline{W} - \frac{1}{4}\min\left\{ \zeta^T\mathbb{E}[S_+ S_+^T \mathbf{1}_{\mathcal{E}^+}]\zeta, \zeta^T\mathbb{E}[S_- S_-^T \mathbf{1}_{\mathcal{E}^-}]\zeta \right\}, \tag{27}
$$

where

$$
\begin{aligned}
& \sup_{\zeta\ge\mathbf{0}} \zeta^T\overline{W} - \tfrac{1}{4}\min\left\{ \zeta^T\mathbb{E}[S_+ S_+^T \mathbf{1}_{\mathcal{E}^+}]\zeta, \zeta^T\mathbb{E}[S_- S_-^T \mathbf{1}_{\mathcal{E}^-}]\zeta \right\} \\
& = \max\left\{ \begin{array}{c} \sup_{\zeta\ge\mathbf{0}} \zeta^T\overline{W} - \tfrac{1}{4}\zeta^T\mathbb{E}[S_+ S_+^T \mathbf{1}_{\mathcal{E}^+}]\zeta, \\ \sup_{\zeta\ge\mathbf{0}} \zeta^T\overline{W} - \tfrac{1}{4}\zeta^T\mathbb{E}[S_- S_-^T \mathbf{1}_{\mathcal{E}^-}]\zeta \end{array} \right\}.
\end{aligned} \tag{28}
$$

Denote

$$
\begin{aligned}
V_+ &= (DM(X_i)'[D(\pi_1 - \pi_0)(X_i)], -\|D(\pi_1 - \pi_0)(X_i)\|^2), \\
V_- &= (DM(X_i)'[D(\pi_1 - \pi_0)(X_i)], \|D(\pi_1 - \pi_0)(X_i)\|^2),
\end{aligned}
$$

then

$$
\begin{aligned}
\mathbf{1}_{\mathcal{E}^+} &= \mathbf{1}\{\zeta^T V_+ \ge 0\}, \\
\mathbf{1}_{\mathcal{E}^-} &= \mathbf{1}\{\zeta^T V_- < 0\}.
\end{aligned}
$$

Let $\zeta_+^*$ satisfy to (29)

$$
\zeta_+^* = \max\left\{ 2\mathbb{E}\left[S_+ S_+^T \mathbf{1}\{\zeta_+^{*T} V_+ \ge 0\}\right]^{-1}\overline{W}, 0 \right\} \tag{29}
$$

and let $\zeta_-^*$ satisfy (30)

$$
\zeta_-^* = \max\left\{ 2\mathbb{E}\left[S_- S_-^T \mathbf{1}\{\zeta_-^{*T} V_- < 0\}\right]^{-1}\overline{W}, 0 \right\}. \tag{30}
$$

Thus

$$
\begin{aligned}
& \sup_{\zeta\ge\mathbf{0}} \zeta^T\overline{W} - \tfrac{1}{4}\zeta^T\mathbb{E}[S_+ S_+^T \mathbf{1}_{\mathcal{E}^+}]\zeta \\
& = \max\left\{ \zeta_+^{*T}\overline{W} - \tfrac{1}{4}\zeta_+^{*T}\mathbb{E}[S_+ S_+^T \mathbf{1}_{\mathcal{E}^+}]\zeta_+^*, 0 \right\} \\
& = \overline{W}^T \mathbb{E}\left[S_+ S_+^T \mathbf{1}\{\zeta_+^{*T} V_+ \ge 0\}\right]^{-1}\overline{W}\mathbf{1}\{\overline{W}\ge 0\},
\end{aligned} \tag{31}
$$

and

$$
\begin{aligned}
& \sup_{\zeta\ge\mathbf{0}} \zeta^T\overline{W} - \tfrac{1}{4}\zeta^T\mathbb{E}[S_- S_-^T \mathbf{1}_{\mathcal{E}^-}]\zeta \\
& = \max\left\{ \zeta_-^{*T}\overline{W} - \tfrac{1}{4}\zeta_-^{*T}\mathbb{E}[S_- S_-^T \mathbf{1}_{\mathcal{E}^-}]\zeta_-^*, 0 \right\} \\
& = \overline{W}^T \mathbb{E}\left[S_- S_-^T \mathbf{1}\{\zeta_-^{*T} V_- \ge 0\}\right]^{-1}\overline{W}\mathbf{1}\{\overline{W}\ge 0\}.
\end{aligned} \tag{32}
$$

Hence by (27) and (28), we have

$$
N\mathcal{R}_{r,\epsilon}(\hat{\mathbb{P}}_N) \lesssim_D \max\left\{ \begin{array}{c} \overline{W}^T\mathbb{E}\left[S_+ S_+^T \mathbf{1}\{\zeta_+^{*T} V_+ \ge 0\}\right]^{-1}\overline{W}, \\ \overline{W}^T\mathbb{E}\left[S_- S_-^T \mathbf{1}\{\zeta_-^{*T} V_- \ge 0\}\right]^{-1}\overline{W} \end{array} \right\}\mathbf{1}\{\overline{W}\ge 0\}
$$

where

$$
\begin{aligned}
V_+ &= (DM(X_i)'[D(\pi_1 - \pi_0)(X_i)], -\|D(\pi_1 - \pi_0)(X_i)\|^2), \\
V_- &= (DM(X_i)'[D(\pi_1 - \pi_0)(X_i)], \|D(\pi_1 - \pi_0)(X_i)\|^2),
\end{aligned}
$$

and $\zeta_+^*, \zeta_-^*$ are defined as in (29), (30).

# C  EXTENSIONS

## C.1  MORE GENERAL APPROXIMATE FAIRNESS PROJECTION DISTANCE

The proposed utility-constrained approximate fairness projection distance can be extended to more generalized formulations via wasserstein projection for group fairness. Let $\hat{\mathbb{P}} \in \mathcal{P}(\mathcal{X})$ be a reference probability measure, $F(\cdot)$ be a convex functional defined on $\mathcal{P}(\mathcal{X})$, $R(\cdot, a)$ be the utility function for sensitivity group $a$. The projection distance is defined as follows:

$$\mathcal{D}_\epsilon^r(\hat{\mathbb{P}}) = \begin{cases} \inf_{\mathbb{Q} \in \mathcal{P}(\mathcal{X})} & \mathcal{W}_c(\mathbb{Q}, \hat{\mathbb{P}})^2 \\ \text{s.t.} & F(\mathbb{Q}) \leq \epsilon \\ & \mathbb{E}_{\mathbb{Q}}[\sum_{a \in \mathcal{S}} p_a(X)\mu(X, a)] \geq r. \end{cases} \tag{33}$$

Suppose $\mathbb{Q}_1 \stackrel{d}{=} \pi_1(X)$, $\mathbb{Q}_0 \stackrel{d}{=} \pi_0(X)$, $X \sim \mathbb{Q}$. Our previously proposed fairness evaluation framework corresponds to the case where $F(\mathbb{Q}) = \mathbb{E}_{\mathbb{Q}}[|\pi_1(X) - \pi_0(X)|]$ according to Lemma B.3. We provide more examples of convex functional $F(\cdot)$ related to the fairness constraints $F(\mathbb{Q}) \leq \epsilon$.

**Example 1** (KL-divergence fairness criterion). *Consider the KL-divergence fairness constraint $\mathrm{D}_{KL}(\mathbb{Q}_1 || \mathbb{Q}_0) \leq \epsilon$, where $\mathrm{D}_{KL}(\mathbb{Q}_1 || \mathbb{Q}_0) := \int_{\mathcal{X}} \pi_1(x) \log(\pi_1(x)/\pi_0(x))\mathbb{Q}(dx)$, which is linear in $\mathbb{Q}$, so $\mathrm{D}_{KL}(\mathbb{Q}_1 || \mathbb{Q}_0)$ is convex in $\mathbb{Q}$.*

**Example 2** (Total-variation fairness criterion). *For the total-variation fairness constraint*

$$\mathrm{TV}(\mathbb{Q}_1, \mathbb{Q}_0) = \sup_{\mathcal{S} \in \mathcal{P}([0,1])} |\mathbb{Q}(\pi_1(X) \in \mathcal{S}) - \mathbb{Q}(\pi_0(X) \in \mathcal{S})| \leq \epsilon.$$

*Note that*

$$|\mathbb{Q}(\pi_1(X) \in \mathcal{S}) - \mathbb{Q}(\pi_0(X) \in \mathcal{S})| = |\mathbb{E}_{\mathbb{Q}}[\mathbf{1}\{\pi_1(X) \in \mathcal{S}\} - \mathbf{1}\{\pi_0(X) \in \mathcal{S}\}]|,$$

*which is convex in $\mathbb{Q}$. Since the supremum of a family of convex function is still convex, the total-variation fairness constraint is convex in $\mathbb{Q}$.*

**Example 3** (Integral Probability Metrics fairness criterion). *For a set of real valued functions $\mathcal{F}$ on $\mathbb{R}^d$, the Integral Probability Metrics (IPM) is defined as*

$$\mathrm{IPM}(\mu, \nu) = \sup_{f \in \mathcal{F}} \int_{\mathbb{R}^d} f d\mu - \int_{\mathbb{R}^d} f d\nu.$$

*One example is $\mathcal{F} = \{f : \|f\|_H \leq 1\}$ where $H$ is a reproducing kernel hilbert space (RKHS), which gives the Maximum Mean Discrepancy (MMD). So*

$$\begin{aligned} \mathrm{IPM}(\pi_1(X), \pi_0(X)) &= \sup_{f \in \mathcal{F}} \int_{\mathbb{R}^d} [f(\pi_1(x)) - f(\pi_0(x))]\mathbb{Q}(dx) \\ &= \sup_{f \in \mathcal{F}} \mathbb{E}_{\mathbb{Q}}[f(\pi_1(X)) - f(\pi_0(X))], \end{aligned}$$

*which is the supremum of a family of linear functions in $\mathbb{Q}$, thus $\mathrm{IPM}(\pi_1(X), \pi_0(X))$ is convex in $\mathbb{Q}$.*

Following this evaluation framework, we can extend the approach outlined above to derive strong duality results, deriving the limiting behavior of test statistics, and implement hypothesis tests. Table 1 summarizes the extensions to other for utility-constrained approximate fairness criteria.

## C.2  MULTIPLE SENSITIVE ATTRIBUTES AND MULTI-LEVEL OR CONTINUOUS TREATMENTS

To extend our setting to $T$-level treatments with multiple sensitive attributes $\mathcal{S}$, with $W_i \in \mathcal{T} = \{0, 1, 2, \ldots, T-1\}$, under confoundedness assumption — $\{Y_i(0), \ldots, Y_i(T-1)\} \perp\!\!\!\perp W_i | X_i$, the expected utility constraint with threshold $r$ is equal to

$$\sum_{a \in \mathcal{S}} \sum_{t \in \mathcal{T}} \mathbb{E}\left[m_t(X_i, a)\pi_{a,t}(X_i)p_a(X_i)\right] \geq r, \tag{34}$$

where $\pi_{a,t}(x) = \mathbb{P}(W_i = t | X_i = x, S_i = a)$, and the $\epsilon$-approximate SDP is defined as

$$\mathbb{E}_{\tau \sim \mathrm{Unif}[0,1]} |\mathbb{Q}(\pi_{a,t}(X_i) > \tau) - \mathbb{Q}(\pi_{a',t}(X_i) > \tau)| \leq \epsilon, \ \forall a, a' \in \mathcal{S}, \ t \in \mathcal{T}. \tag{35}$$

Table 1: Extension to more general approximate fairness projection distances

| Utility constraint | Approximate fairness criterion | $\epsilon$-approximate fairness constraint |
|---|---|---|
| | $\epsilon$-approximate DP | $\left|\mathbb{E}_{\mathbb{Q}}[\pi_1(X)] - \mathbb{E}_{\mathbb{Q}}[\pi_0(X)]\right| \leq \epsilon^1$ |
| $\mathbb{E}[Y(W)] \geq r$ | $\epsilon$-approximate KL-divergence fairness | $\mathbb{E}_{\mathbb{Q}}\left[\pi_1(X) \log\left(\frac{\pi_1(X)}{\pi_0(X)}\right)\right] \leq \epsilon$ |
| | $\epsilon$-approximate TV fairness | $\left|\mathbb{E}_{\mathbb{Q}}\left[\mathbf{1}\{\pi_1(X) \in \mathcal{S}\} - \mathbf{1}\{\pi_0(X) \in \mathcal{S}\}\right]\right| \leq \epsilon$ |
| | $\epsilon$-approximate IPM fairness | $\text{IPM}\big(\pi_1(X), \pi_0(X)\big) \leq \epsilon$ |

We replace the constraints of (P) with (34) and (35).

To extend our setting to continuous treatments $\mathcal{T} \subset \mathbb{R}$, we study infinitesimal interventions on the treatment level motivated by the work of Powell et al. (1989), and the expected utility of such intervention is defined as

$$\left[\frac{d}{d\nu}\mathbb{E}\left[Y_i(W_i + \nu I(X_i, S_i))\right]\right]_{\nu=0},$$

where $I : \mathcal{X} \times \mathcal{S} \in \{0, 1\}$ is a binary function representing the treatment policy according to the given contexts. Let $m(w, x, a) = \mathbb{E}[Y_i(w)|X_i = x, S_i = a]$. Under unconfoundedness assumption $\{Y_i(w)\}_{w \in \mathcal{T}} \perp\!\!\!\perp W_i | X_i, S_i$ and that $\{Y_i(w)\}_{w \in \mathcal{T}}$ are uniformly bounded by a constant, we have

$$\mathbb{E}\left[Y_i(W_i + \nu I(X_i, S_i))\right] = \mathbb{E}\left\{\int_{w \in \mathcal{T}} \mathbb{E}\left[Y_i(w + \nu I(X_i, S_i))|X_i, S_i\right] \pi(w|X_i, S_i)dw\right\}$$

$$= \mathbb{E}\left[\int_{w \in \mathcal{T}} m(w + \nu I(X_i, S_i)), X_i, S_i)\pi(w|X_i, S_i)dw\right]$$

$$= \sum_{a \in \mathcal{S}} \int_{w \in \mathcal{T}} \mathbb{E}\left[m(w + \nu I(X_i, a)), X_i, a)\pi(w|X_i, a)p_a(X_i)\right] dw.$$

where the integral and the expectations are exchangeable above by using Fubini Theorem as a result of the uniform boundedness of the potential outcomes. Then under some additional regularity conditions, we can exchange the derivative (with respect to $\nu$) with the integrals and the expectations, so that

$$\frac{d}{d\nu}\mathbb{E}\left[Y_i(W_i + \nu I(X_i, S_i))\right]_{\nu=0}$$

$$= \sum_{a \in \mathcal{S}} \int_{w \in \mathcal{T}} \mathbb{E}\left[D_w m(w, X_i, a)I(X_i, a)\pi(w|X_i, a)p_a(X_i)\right] dw,$$

where $D_w m$ is the gradient of $m$ taken with respect to $w$. The utility constraint is defined as

$$\sum_{a \in \mathcal{S}} \int_{w \in \mathcal{T}} \mathbb{E}\left[D_w m(w, X_i, a)I(X_i, a)\pi(w|X_i, a)p_a(X_i)\right] dw \geq r. \tag{36}$$

Define

$$\Pi(X_i, a) := I(X_i, a) \int_{w \in \mathcal{T}} \pi(w|X_i, a)dw,$$

the $\epsilon$-approximate SDP is defined as

$$\mathbb{E}_{\tau \sim \text{Unif}[0,1]} \left|\mathbb{Q}(\Pi(X_i, a) > \tau) - \mathbb{Q}(\Pi(X_i, a') > \tau)\right| \leq \epsilon, \ \forall a, a' \in \mathcal{S}, \ t \in \mathcal{T}. \tag{37}$$

where $\Pi(X_i, a)$ captures the interaction between the average pre-intervention treatment level and the binary intervention. Then we replace the constraints of (P) with (36) and (37) under the setting with continuous treatment and multiple sensitive attributes.

In both extended cases, the expectations of the constraints are taken with respect to the distribution of $X_i$. Thus, the formality of the hypothesis testing framework and the Wasserstein projection distance remain unchanged, and the proof techniques for the setting with binary treatments and binary sensitive attributes apply directly once the necessary additional regularity conditions are imposed.

## D   ON EXTENDING EMPIRICAL STUDIES TO UNSTRUCTURED DATA

Beyond the structured-data applications examined in the main text, our framework naturally extends to unstructured domains such as natural language processing (NLP), computer vision, and recommender systems. Given the complexity of these tasks and the primarily theoretical focus of our work, we provide only a high-level discussion of how our hypothesis test could be applied, leaving detailed empirical investigations to future research. These extensions illustrate how the choice of $(\epsilon, r)$ adapts to different empirical contexts—accuracy in NLP, diagnostic benefit in imaging, and engagement in recommendations—while our test offers a unified approach to evaluating fairness–utility trade-offs.

**NLP data (Resume Screening).**   In text-based classification tasks such as resume screening, datasets like Bias in Bios link occupation labels with gender. Here, utility $r$ can be defined as maintaining predictive accuracy above a threshold, while fairness tolerance $\epsilon$ limits group disparities in predicted selection rates across thresholds. Fine-tuning a language model (e.g., BERT) and applying our test allows one to assess whether observed gender gaps are systematic or due to randomness.

**Medical Imaging (Skin Cancer Detection).**   Datasets such as **Fitzpatrick17k** with skin-tone annotations can be paired with melanoma classification data. Utility $r$ corresponds to minimum diagnostic accuracy (e.g., sensitivity), while $\epsilon$ controls disparities in screening probabilities across skin tones. Training a CNN and applying our procedure provides a test of whether differences in outcomes reflect structural bias or noise.

**Recommender Systems (MovieLens).**   In recommendation platforms, datasets like MovieLens enable analysis of exposure disparities across gender or age groups. Here, $r$ reflects minimum engagement or rating accuracy, and $\epsilon$ bounds disparities in recommendation probabilities. Applying our test to collaborative filtering models helps determine whether unequal exposure is intrinsic to the system or explained by sampling variation.

## E   DATASET DESCRIPTIONS

**COMPAS** dataset. The COMPAS (*Correctional Offender Management Profiling for Alternative Sanctions*) dataset a widely adopted commercial tool that assists judges and parole officers in algorithmically predicting a defendant's recidivism risk. The dataset comprises criminal records from a two-year follow-up period post-sentencing. For our fairness analysis, sex serves as the sensitive attribute.

**Arrhythmia** dataset. Arrhythmia is from UCI repository, where the aim of this data set is to distinguish between the presence and absence of cardiac arrhythmia and classify it in one of the 16 groups. The dataset consists of 452 samples and we use the first 12 features among which the gender is the sensitive feature. For our purpose, we construct binary labels between 'class 01' ('normal') and all other classes (different classes of arrhythmia and unclassified ones).

**Drug** dataset. The Drug dataset contains answers of 1885 participants on their use of 17 legal and illegal drugs. We concern the cannabis usage as a binary problem, where the label is 'Never used' VS 'Others' ('used'). There are 12 features including age, gender, education, country, ethnicity, NEO-FFI-R measurements, impulsiveness measured by BIS-11 and sensation seeking measured by ImpSS. Among those, we choose ethnicity (black vs others) as the sensitive attribute.

## F   ADDITIONAL RESULTS FOR NUMERICAL EXPERIMENTS

The utility function for the empirical study is defined as follows. Let $\mathcal{D} = \{(X_i, A_i, W_i)\}_{i=1}^{N}$ be the testing dataset, where $W_i \in \{0, 1\}$ is the ground truth label. For each sample, the utility function is defined as the expected concordance between the predicted label distribution and the observed outcome: $Y_i = \pi(X_i, A_i)W_i + (1 - \pi(X_i, A_i))(1 - W_i)$ where $\pi(X_i, A_i) \in [0, 1]$ denotes $\mathbb{P}(W_i = 1 \mid X_i, A_i)$. For logistic regression, the probability is given by the sigmoid transformation: $\pi_{\text{LR}}(X_i, A_i) = \frac{1}{1+\exp(-\theta^T(X_i, A_i))}$. For support vector machines with Platt scaling, the probability estimate is $\pi_{\text{SVM}}(X_i, A_i) = \frac{1}{1+\exp(\alpha \cdot f(X_i, A_i) + \beta)}$ where $f(X_i, A_i) = \theta^T(X_i, A_i) + b$.

Figure 3: Empirical study results: comparisons for rejection rates of fairness hypothesis under different values of $(\epsilon, r)$ for all three real data sets. The third column shows the heatmap plot for the difference of rejection rates between bootstrap test and our method. Whenever our test rejects, the bootstrap test also rejects; moreover, the bootstrap test rejects in additional cases where our method does not. Thus under all values of $(\epsilon, r)$, the naive bootstrap is consistently more conservative than the Wasserstein projection test.

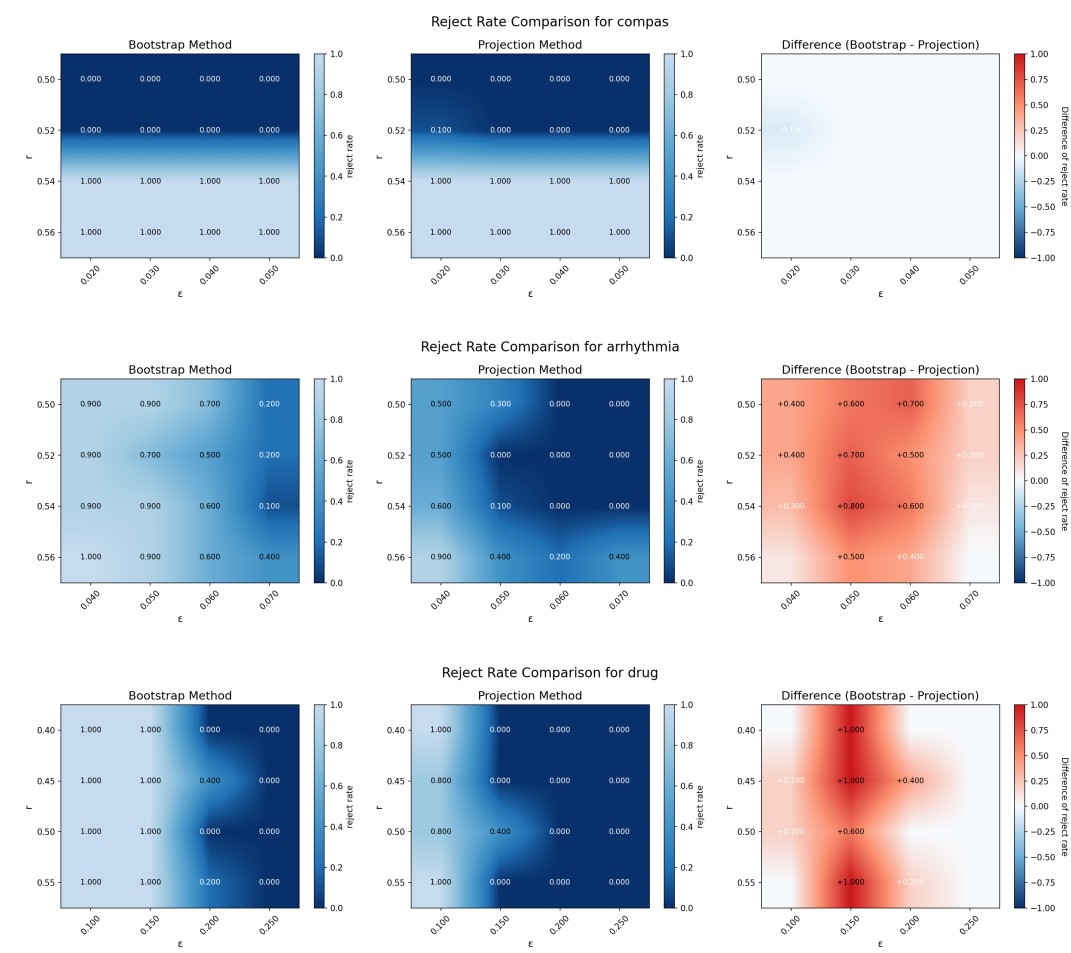

Table 2: Empirical study results: reject rate comparison across all three datasets

| COMPAS | | | | Arrhythmia | | | | Drug | | | |
|---|---|---|---|---|---|---|---|---|---|---|---|
| $\epsilon$ | $r$ | Proj. | Boot. | $\epsilon$ | $r$ | Proj. | Boot. | $\epsilon$ | $r$ | Proj. | Boot. |
| 0.02 | 0.50 | 0.0 | 0.0 | 0.04 | 0.50 | 0.5 | 0.9 | 0.10 | 0.40 | 1.0 | 1.0 |
| 0.02 | 0.52 | 0.1 | 0.0 | 0.04 | 0.52 | 0.5 | 0.9 | 0.10 | 0.45 | 0.8 | 1.0 |
| 0.02 | 0.54 | 1.0 | 1.0 | 0.04 | 0.54 | 0.6 | 0.9 | 0.10 | 0.50 | 0.8 | 1.0 |
| 0.02 | 0.56 | 1.0 | 1.0 | 0.04 | 0.56 | 0.9 | 1.0 | 0.10 | 0.55 | 1.0 | 1.0 |
| 0.03 | 0.50 | 0.0 | 0.0 | 0.05 | 0.50 | 0.3 | 0.9 | 0.15 | 0.40 | 0.0 | 1.0 |
| 0.03 | 0.52 | 0.0 | 0.0 | 0.05 | 0.52 | 0.0 | 0.7 | 0.15 | 0.45 | 0.0 | 1.0 |
| 0.03 | 0.54 | 1.0 | 1.0 | 0.05 | 0.54 | 0.1 | 0.9 | 0.15 | 0.50 | 0.4 | 1.0 |
| 0.03 | 0.56 | 1.0 | 1.0 | 0.05 | 0.56 | 0.4 | 0.9 | 0.15 | 0.55 | 0.0 | 1.0 |
| 0.04 | 0.50 | 0.0 | 0.0 | 0.06 | 0.50 | 0.0 | 0.7 | 0.20 | 0.40 | 0.0 | 0.0 |
| 0.04 | 0.52 | 0.0 | 0.0 | 0.06 | 0.52 | 0.0 | 0.5 | 0.20 | 0.45 | 0.0 | 0.4 |
| 0.04 | 0.54 | 1.0 | 1.0 | 0.06 | 0.54 | 0.0 | 0.6 | 0.20 | 0.50 | 0.0 | 0.0 |
| 0.04 | 0.56 | 1.0 | 1.0 | 0.06 | 0.56 | 0.2 | 0.6 | 0.20 | 0.55 | 0.0 | 0.2 |
| 0.05 | 0.50 | 0.0 | 0.0 | 0.07 | 0.50 | 0.0 | 0.2 | 0.25 | 0.40 | 0.0 | 0.0 |
| 0.05 | 0.52 | 0.0 | 0.0 | 0.07 | 0.52 | 0.0 | 0.2 | 0.25 | 0.45 | 0.0 | 0.0 |
| 0.05 | 0.54 | 1.0 | 1.0 | 0.07 | 0.54 | 0.0 | 0.1 | 0.25 | 0.50 | 0.0 | 0.0 |
| 0.05 | 0.56 | 1.0 | 1.0 | 0.07 | 0.56 | 0.4 | 0.4 | 0.25 | 0.55 | 0.0 | 0.0 |

Figure 4: Empirical study results. Top row: fairness–utility tradeoff. Bottom row: rejection rates. The parameters are $r = 0.5, \epsilon = 0.02$ for COMPAS, $r = 0.5, \epsilon = 0.04$ for Arrhythmia, and $r = 0.5, \theta = 0.2$ for Drug. Our observations indicate that stronger regularization leads to an increase in the $0.95$ quantile of the stochastic upper bound and a lower likelihood of rejecting the null hypothesis—i.e., concluding that the policy is unfair.

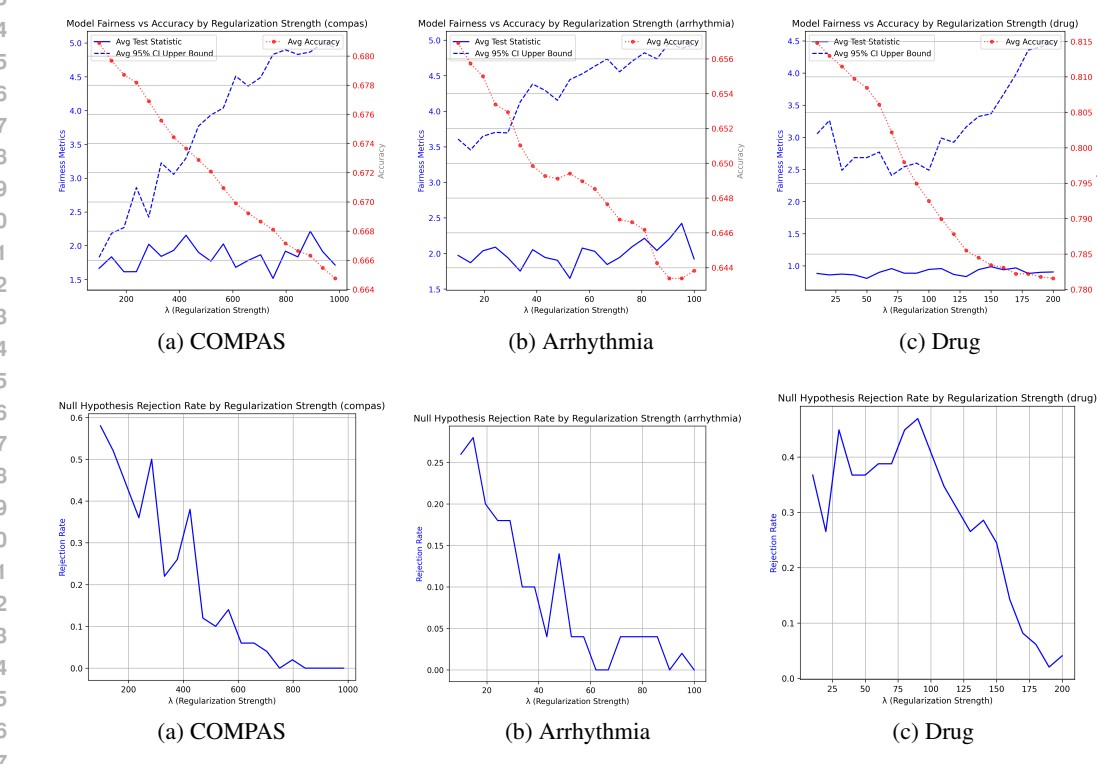

(a) COMPAS     (b) Arrhythmia     (c) Drug

(a) COMPAS     (b) Arrhythmia     (c) Drug

Figure 5: Simulated data results: comparisons for rejection rates of fairness hypothesis under different values of $(\epsilon, r)$. The third column shows the heatmap plot for the difference of rejection rates between bootstrap test and our method. Whenever our test rejects, the bootstrap test also rejects; moreover, the bootstrap test rejects in additional cases where our method does not. Thus under all values of $(\epsilon, r)$, the naive bootstrap is consistently more conservative than the Wasserstein projection test.

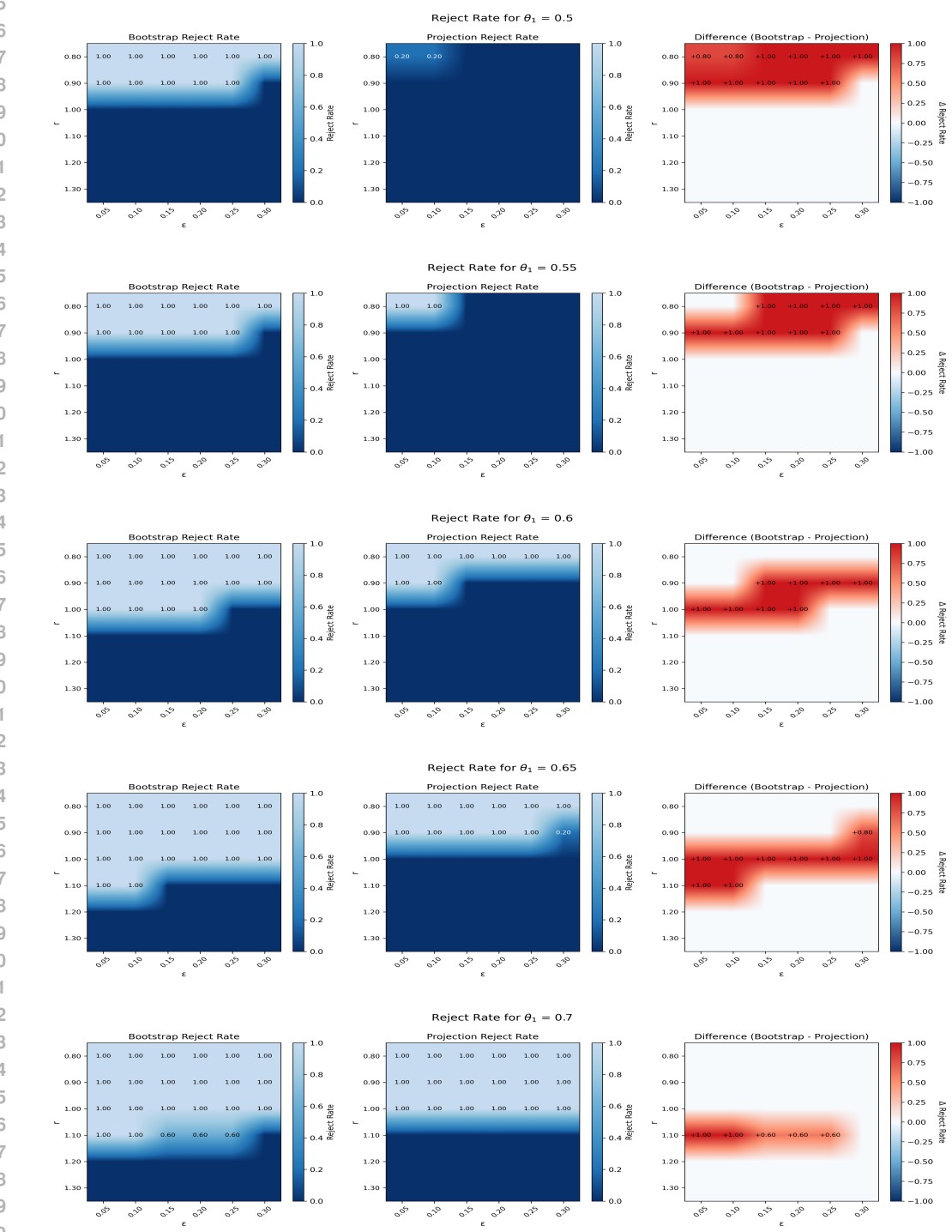

