# OpenReview forum: "Testing Fairness with Utility Trade-offs: A Wasserstein Projection Approach"
_ICLR.cc/2026/Conference — Submitted to ICLR 2026_

### Official Review · Reviewer_yZVT · 2025-10-15

**Soundness:** 3
**Presentation:** 3
**Contribution:** 2
**Rating:** 4
**Confidence:** 3

**Summary:**

The paper builds a statistical hypothesis testing framework to check whether a model is approximately fair in terms of SDP, while still results in enough utility. It defines a null set of fair and utility-ensured distributions $\mathcal{g}(r,\epsilon)$ and then performs a hypothesis testing to examine how far the empirical data distribution is from this set using Wasserstein projection. If this distance is large, the test rejects the null hypothesis.

**Strengths:**

1. The framework is theoretically rigorous. The authors develop formal hypothesis testing procedures, prove strong duality/asymptotic results, and demonstrate clear statistical guarantees for the proposed test.

2. The problem formulation is concrete and well-structured.

3. Empirical evaluations are provided.

**Weaknesses:**

1. While the framework is theoretically rigorous, I am not sure of the practical significance, i.e., the test indicates whether "fairness–utility trade-offs" are significant, but it seems not to offer insight into how such findings inform model design or real-world decision-making.

2. The work mainly contributes a statistical testing procedure rather than a learning algorithm. This makes me a bit unsure about its fit for ICLR.

**Questions:**

How can practitioners use the framework?
How can we use the framework to guide learning algorithm designs?

---

> ### Author Response · Authors · 2025-11-17
>
> We thank the reviewer for the comments.
>
> (W1) As explained in the paper, our hypothesis-testing framework is not merely descriptive; by allowing stakeholders to set a utility threshold $r$ and a fairness slack $\epsilon$, it provides operative metrics that directly inform model deployment decisions.
>
> For example, if the test rejects the null, then one knows that given current model/policy and the specified (r, ε) the fairness–utility pair is infeasible. This immediately signals that one must either relax the fairness requirement (increase ε), lower the utility target (reduce r), or redesign the policy — all of which are concrete design or governance manoeuvres.
>
> On the other hand, accepting the null gives the decision-maker statistical assurance that the policy is consistent with the desired fairness–utility specification. We further illustrate how tuning regularization (in Section 4) moves the model along the frontier of feasible (r, ε) pairs.
>
> In short, while our framework does not automatically prescribe swaps of features or algorithm types, it clearly links audit outcomes to model design/governance choices, making it a practical tool for real-world decision-making. We will emphasise this connection more strongly in the paper to clarify the path from test result → design action.
>
> ---
> (W2): While our contribution indeed takes the form of a statistical testing framework, it is specifically designed for and motivated by machine-learning systems. The proposed Wasserstein-projection test addresses a central ML question — assessing fairness–utility trade-offs in model outputs — and thus directly supports algorithmic evaluation and design.
>
> Our method goes beyond a generic statistical procedure:
>
> - It introduces a new composite hypothesis tailored to fairness-aware ML (approximate group fairness + utility threshold).
>
> - It establishes strong duality and asymptotic results connecting optimal-transport geometry with fairness evaluation.
>
> - It is practically integrated into ML pipelines as an auditing tool applicable to classifiers such as logistic regression and SVMs.
>
> We view this as a valuable contribution that bridges statistical inference and ML fairness auditing — an area explicitly within ICLR’s scope (“theoretical understanding and societal aspects of representation learning”). Moreover, the framework lays conceptual groundwork for developing future fairness-aware learning algorithms inspired by Wasserstein projections, especially considering utility-fairness tradeoff, which is under-explored in exsiting literature.
>
> We will clarify this ML alignment and impact in the paper.
>
> ---
> Please let us know if our response addresses your concerns. If not, we’ll be glad to answer any further questions you may have.

---

> > ### Comment · Reviewer_yZVT · 2025-11-20
> >
> > Thanks for your response. I agree that the paper has some technical contribution, and the framework can test whether the fairness-utility tradeoff actually exists. However, as I mentioned, I will be more excited to see the contribution to ML model.
> >
> > Nevertheless, I still regard this paper as having some merit and would not mind it being accepted to ICLR. But currently, I will keep my score.

---

> ### Author Response · Authors · 2025-11-28
> **Our contribution**
>
> We thank the reviewer for recognizing our work's theoretical contribution.
>
> In addition, we would like to emphasize that, to the best of our knowledge, the only existing works on fairness hypothesis testing via Wasserstein projection are Taskesen et al. (2021), Si et al. (2021)—both cited in line 211—and one unpublished arXiv preprint [1]. Our paper is, to our knowledge, the first to implement a hypothesis test of fairness under a fairness–utility trade-off framework using the Wasserstein projection method. Furthermore, our real-data experiments follow the empirical settings adopted in the most closely related works (Taskesen et al., 2021; Si et al., 2021), and our empirical results meaningfully support our theoretical fairness-testing framework by illustrating the inherent utility trade-offs.
>
> For these reasons, we therefore believe our paper offers significant value to the ICLR community on fairness literature.
>
> [1] Li, W., Park, Y. P., \& Duc, K. D. (2025). Wasserstein projection distance for fairness testing of regression models. arXiv preprint arXiv:2510.04114.

---

### Official Review · Reviewer_Yb7p · 2025-10-18

**Soundness:** 3
**Presentation:** 3
**Contribution:** 2
**Rating:** 6
**Confidence:** 4

**Summary:**

This paper studies statistical fairness testing under a joint fairness–utility perspective. The authors propose a Wasserstein projection-based test that simultaneously accounts for (i) approximate fairness constraints, formalized as a relaxed variant of Strong Demographic Parity (SDP) that must hold across all thresholds; and (ii) a minimum-utility requirement under a potential-outcomes model. The idea is to project the empirical distribution onto the set of distributions satisfying both constraints and to measure the resulting Wasserstein distance; this projection defines a test statistic with strong duality guarantees. The paper derives asymptotic properties for the test statistic (level-$\alpha$ control, convergence) and provides a convex dual program that can be efficiently computed. Experiments on synthetic and real datasets illustrate how varying the fairness tolerance or utility constraint shifts rejection rates, demonstrating interpretability of fairness–utility trade-offs.

**Strengths:**

echnical soundness and mathematical rigor.
The paper is well executed on the theoretical side: the Wasserstein projection formulation, convex duality derivation, and asymptotic control of the test statistic are all clearly presented and appear correct. The proofs are detailed and internally consistent, suggesting a high level of mathematical care.

Conceptual clarity of fairness–utility trade-off.
Although not new in spirit, the joint treatment of fairness and utility within a single optimization constraint provides a structured way to visualize and quantify trade-offs. The use of Wasserstein geometry offers interpretability through distances in distributional space.

Quality of exposition.
Writing is clean and formal. Definitions, lemmas, and proofs are well organized, and the authors use intuitive diagrams to illustrate projection behavior and rejection regions. The paper reads smoothly and is self-contained.

Practical interpretability.
The fairness–utility trade-off plots and contour diagrams convey useful intuitions for how fairness constraints affect acceptance regions. Even if originality is limited, the framework might help practitioners audit fairness tests in a rigorous statistical way.

**Weaknesses:**

Limited originality relative to prior OT-based fairness work. The main conceptual contribution (casting fairness testing as a Wasserstein projection with dual formulation) is incremental, given several recent studies that already apply optimal transport to fairness measurement or correction (e.g., Gordaliza et al., 2019; Chzhen et al., 2020; Xu et al., 2022; Pérez-Serrano & Oosterlee, 2023). The novelty here lies mainly in combining fairness and utility in one projection, but this extension is modest. Suggestion: Explicitly acknowledge overlapping prior work and clarify what differentiates this method, e.g., whether the test’s dual form or asymptotic control offers unique theoretical guarantees.

Weak empirical evidence. Experiments are limited to small synthetic setups and a simple tabular dataset (COMPAS). There are no comparisons to other fairness-testing metrics or OT-based methods. As a result, the claimed advantages remain speculative.
Suggestion: Include empirical baselines (e.g., equalized-odds distance, Kolmogorov–Smirnov test, prior OT fairness auditors) to substantiate claims.

Unclear practical relevance. The fairness constraint (“approximate strong demographic parity across thresholds”) and the potential-outcomes utility model are abstract and may not map easily to the fairness notions used in ML practice (EO, DP, PPV). This makes it hard for practitioners to adopt the test. Suggestion: Offer a translation table or case study linking this formalism to standard metrics.

Lack of guidance on parameter selection. The fairness and utility tolerances ($\epsilon_f$, $\epsilon_u$) are varied in plots but not justified. Without principled tuning, users cannot reproduce or interpret real-world decisions based on this test. Suggestion: Add heuristics or theoretical discussion for choosing tolerance levels.

Scalability and applicability not demonstrated. Solving the Wasserstein projection exactly via convex optimization may be intractable in high-dimensional data; no experiments beyond low-dimensional settings are shown.
Suggestion: Provide at least one medium-scale experiment or mention possible entropic/Sinkhorn approximations, or sequential transport.

Positioning and related work. The paper’s framing underplays the volume of closely related literature on fairness via optimal transport and statistical testing. Without stronger contextualization, the work risks appearing as a variant rather than a new direction.

**Questions:**

Statistical power: How does the test’s power compare empirically to simpler metrics (e.g., equality-of-odds distance, or KS-distance between groups)?

Utility estimation: In practice, how are potential-outcome utilities estimated? Are counterfactual estimates required, or can surrogate utility functions (like accuracy or F1) suffice?

Choice of $\epsilon$ parameters: Could you provide theoretical or empirical guidance on selecting fairness and utility tolerance levels?

Scalability: Can the proposed projection be implemented using entropic-regularized OT solvers (e.g., Sinkhorn) to handle large datasets?

Generalization beyond binary sensitive attributes: Can the test handle continuous or multi-valued sensitive attributes, or does the constraint structure depend on discrete group partitions?

Connection to fairness metrics: How does approximate SDP relate to other definitions (equalized odds, predictive parity)? Clarifying this would make the test’s scope clearer.

Runtime / convergence: What are typical runtimes of the dual solver, and how sensitive are they to regularization parameters?

Visualization: The fairness–utility contour plots are insightful; adding joint contour density or rejection boundaries for real datasets would strengthen interpretability.

---

> ### Author Response · Authors · 2025-11-18
>
> We thank the reviewer for the comments and the recognition of the contribution of our work.
>
> (W1) Thank you for pointing out the relation of our work to prior OT-based fairness studies. Our contribution is distinct in three key aspects. First, prior work primarily uses OT for fairness measurement or correction, whereas we formulate a statistical test that jointly evaluates fairness and utility through a Wasserstein projection. Second, our framework provides theoretical guarantees via a novel dual representation that allows asymptotic control of the test statistic and principled critical value computation—features absent in many existing OT fairness methods. Third, our formulation is black-box and model-agnostic, enabling fairness auditing across arbitrary predictive systems without retraining or internal access. Together, these aspects elevate the contribution from incremental adaptation to a new class of fairness–utility hypothesis testing tools. We will clarify these distinctions and explicitly acknowledge related OT-based fairness works in the paper.
>
> (W2) We will include empirical baselines to substantiate claims.
>
> (W3) We appreciate the reviewer's suggestion, and will offer a translation table linking this formalism to standard fairness metrics.
>
> (W4) We've  discussed how to choose $(\epsilon,r)$ in lines 303–310 and Appendix F. These choices are typically context-dependent. We would be happy to include further discussion on this point.
>
> (W5) Our main contribution is theoretical—establishing a statistically valid Wasserstein-projection test with strong duality and asymptotic guarantees. While current experiments focus on moderate-dimensional data for clarity, the method is not inherently limited to low-dimensional settings. The Wasserstein projection can be efficiently approximated using entropic regularization (Sinkhorn iterations) or stochastic/sequential transport solvers (Peyré and Cuturi, 2019), which integrate naturally into our framework since only a consistent projection estimator is required for validity. We will add discussion of these scalable variants and highlight that extending to large-scale, high-dimensional settings is a natural and feasible next step.
>
> (W6) We acknowledge prior works using Wasserstein-projection–type hypotheses. However, none incorporate fairness evaluation together with a utility trade-off, which is the key novelty of our work. Our test jointly enforces approximate group fairness and minimum utility constraints, providing both theoretical guarantees (strong duality, asymptotic validity) and practical interpretability for ML auditing. We will clarify this distinction and better contextualize related work in the revised version.
>
> ---
>
> (Q1) If we replace SDP with equality-of-odds (EO) the test mechanics remain the same (the constraint set $\mathcal F$ changes), and our theoretical guarantees (strong duality, asymptotic validity) still apply so long as the EO constraint is expressed in a convex, measurable form. However, the power profile changes: a test built around EO will be more sensitive to alternatives that change conditional error rates (TPR/FPR differences across groups), while a test built around SDP will be more sensitive to alternatives that change marginal decision rates or other population-level shifts. EO constraints require conditioning on $Y$, which can increase variance (and sample complexity) relative to SDP, so in moderate sample sizes an EO-based test may need larger $N$ to achieve the same power for some alternatives.
>
> (Q2) Potential outcome utilities can be estimated by fitting parametric or nonparametric models from data. We would need the utilities be smooth in $x$ such that Assumption 2 (line 319-320) holds.
>
> (Q3) We've provided this empirical guidance between lines 303-309 and Appendix F.
>
> (Q4) Adding entropic regularization would break the strong duality result, and the limiting distribution of the test statistic might no longer hold. However, from a practical perspective, entropic regularization could be beneficial for high-dimensional or large-scale datasets. We therefore leave this promising direction for future work.
>
> (Q5) We've included multi-valued sensitive attributes in Appendix D.2. But SDP is not well defined for continuous sensitive attributes.
>
> (Q6) SDP controls marginal dependence between $W$ and $S$, whereas equalized odds additionally conditions on $Y$, and predictive parity conditions on $W$. Hence, SDP is stricter than EO in marginal independence but does not enforce conditional independence given outcomes.
>
> (Q7) Each empirical datasets runs around 300s, agnostic to regularization parameters.
>
> (Q8) We will add the contour plot for the datasets in the paper.
>
> We appreciate the reviewer's comment and please let us know if you have additional questions or concerns.

---

> > ### Comment · Reviewer_Yb7p · 2025-11-25
> >
> > I thank the authors for their thoughtful and well-organized rebuttal. The clarifications around the Wasserstein projection, the dual formulation, and the interpretation of the approximate-SDP constraint were helpful. I also appreciate the additional explanations about computational tractability and the empirical illustrations of fairness–utility trade-offs.
> >
> > That said, several of the core concerns in my original review remain only partially addressed:
> >
> > 1. Originality relative to prior OT-based fairness work.
> > The rebuttal clarifies the conceptual distinction between this projection and prior OT formulations, but the connection still feels incremental. A large body of recent work applies OT, Wasserstein distance, or projection-based approaches to fairness auditing and correction. The main novelty here is the joint fairness–utility constraint, which is interesting but relatively modest. The response does not provide a clearer positioning that would significantly change my assessment of novelty.
> >
> > 2. Limited empirical support.
> > The added clarifications do not change the fact that the empirical evaluation remains narrow (small synthetic setups and a single tabular dataset). No comparisons to alternative fairness tests or OT-based baselines are provided. As a result, it is still difficult to evaluate statistical power, robustness, or practical relevance.
> >
> > 3. Practicality and scalability.
> > While the authors explain why their convex optimization remains tractable in moderate dimensions, the method’s applicability to real ML settings (e.g., embeddings, large feature spaces, or structured outputs) remains untested. The rebuttal provides no new experiments that would strengthen confidence in scalability.
> >
> > 4. Interpretability for practitioners.
> > Although the authors elaborate on approximate SDP, the mapping from this formal constraint to familiar fairness metrics still feels abstract. This limits the accessibility and real-world usability of the test.
> >
> > Overall, the rebuttal improves clarity but does not substantially shift my view of the work’s novelty, empirical grounding, or practical impact. I therefore maintain my original evaluation.

---

> ### Author Response · Authors · 2025-11-27
> **Our responses to reviewer Yb7p's additional concerns, and additional results we added in the paper**
>
> We sincerely thank the reviewer for the response and for the recognition of our work's theoretical contribution.
>
> **Regarding reviewer's remaining concerns:**
>
> **(1) point 1 of reviewer's remaining concern:**
>
> We would like to emphasize that, to the best of our knowledge, the only existing works on fairness hypothesis testing via Wasserstein projection are Taskesen et al. (2021), Si et al. (2021)—both cited in line 211—and one unpublished arXiv preprint [1]. Our paper is, to our knowledge, the first to implement a hypothesis test of fairness under a fairness–utility trade-off framework using the Wasserstein projection method. Furthermore, our real-data experiments follow the same empirical settings adopted in the most closely related works (Taskesen et al., 2021; Si et al., 2021). We therefore believe that our work offers novel contributions to this line of research, and that our empirical results meaningfully support our theoretical fairness-testing framework by illustrating the inherent utility trade-offs.
>
> [1] Li, W., Park, Y. P., \& Duc, K. D. (2025). Wasserstein projection distance for fairness testing of regression models. arXiv preprint arXiv:2510.04114.
>
> ---
>
> **(2) empirical baselines & point 2 of reviewer's remaining concern:**
>
> - We have provided empirical baselines comparing with bootstrap tests for all numerical studies, showing the validity and usefulness of our method, in our last response to reviewer NFVQ.
>
> - The baselines suggested by the reviewer are fairness metrics that omit utility trade-offs, and thus a direct comparison is not meaningful: these metrics and our hypothesis test target fundamentally different goals. Appendix C provides a more general formulation of approximate fairness–projection distances, under which one could extend the fairness constraint beyond approximate SDP to metrics such as EO distance. However, this extension would necessitate re-establishing strong duality and re-deriving the asymptotic distribution, which lies outside the intended scope of our paper.
>
> ---
>
> **(3) point 3 of reviewer's remaining concern:**
>
> As we replied in our response to reviewer NFVQ, our real-data experiments follow the same empirical settings as the most closely related works (Taskesen et al., 2021; Si et al., 2021).
>
> In addition, we conducted supplementary experiments comparing our method with a naive bootstrap test on both simulated and real datasets, further supporting the validity of our approach.
>
> Third, this paper is primarily a theoretical contribution; the empirical and numerical studies are included to verify the theoretical results, which they successfully do. We respectfully disagree that running experiments on substantially larger datasets is a major concern for a theoretical paper, and we respectfully don't think this should diminish the recognition of our contribution.
>
> ---
>
> **(4) point 4 of reviewer's remaining concern:**
>
> We have already provided explanations in the original submission to enhance interpretability for practitioners. In particular, the interpretation of the fairness metrics under the utility trade-off is explained in detail through an illustrative example in lines 307–315 of the most up-to-date version of the paper.
>
> ---
>
> **Additional points regarding the earliest comment by the reviewer (the additional translation table and plots we have promised to add in the earlier comment):**
>
> - We have added a translation table (Table 1) in Appendix C in the revised version of our paper.
>
> - We have also provided contour plot with heatmap for the rejection rates under different values of $(\epsilon,r)$, for both our Wasserstein projection test and naive bootstrap test in Appendix F in the revised version of our paper, and we have also provided detailed discussion on this in the paragraph ``Comparison with Naive Bootstrap Tests'' on page 9 of the revised version of our paper.

---

### Official Review · Reviewer_uyZP · 2025-10-22

**Soundness:** 3
**Presentation:** 1
**Contribution:** 3
**Rating:** 4
**Confidence:** 3

**Summary:**

This paper presents a statistical hypothesis testing framework that jointly evaluates approximate fairness and utility in algorithmic decision-making. The proposed test assesses whether a data distribution satisfies approximate Strong Demographic Parity (for fairness) and exhibits low expected utility loss (for efficiency), using observed classification labels and predicted propensity scores. The method is based on a Wasserstein projection of the data to the nearest distribution satisfying both criteria, with an asymptotic stochastic upper bound on the projection distance. The main results address the binary case, with generalizations to the multi-valued case discussed in Appendix D.

**Strengths:**

The strengths of the paper come from its contributions, summarized at the end of Section 1.2 (lines 99-105). In particular, it proposes a statistical test that is computationally tractable, interpretable, and broadly applicable. The idea of mapping the data to the nearest $r$-efficient and $\epsilon$-fair distribution using a Wasserstein projection appears to be novel. The mathematical derivations appear technically sound based on an initial reading of the appendix. Based on the examples provided in Section 4 and Appendix F, along with the discussion of extensions beyond the binary case in Appendix D, the test seems to be widely applicable in practice.

**Weaknesses:**

There are several significant aspects of the paper’s presentation that could be improved, particularly with respect to clarity and reproducibility. The primary weakness lies in the limited intuition provided for the main result, Theorem 3.2. Specifically, the connection between the right-hand side of Equation (8) and the test statistic from Hypothesis Test (4) is not clearly explained. Readers are left to infer how $\eta_{1-\alpha}$ corresponds to the $(1-\alpha)$ quantile of the distribution in (8), and how Equations (7) and (P) justify rejecting $\mathcal{H}_0$ when $t_N(\epsilon, r) > \eta_{1-\alpha}$ (lines 374–377). The method for computing the test statistic itself (lines 392–398) is similarly vague.

Furthermore, while the numerical experiments offer helpful illustrations, the paper lacks a clear description of how to implement the proposed test in practice. In particular, it would benefit from a worked example using real-world data mapped into the framework introduced in Equation (3), as well as guidance on how the approach generalizes beyond the binary case. As written, the current exposition makes it difficult for others to reproduce or build upon the method. With significant revisions to improve exposition and accessibility, this paper could make a valuable contribution.

Below is a list of specific points that could be communicated more clearly:
- The infinity signs on lines 213--215 are undefined
- Gradients $\nabla$ and $D$ are used interchangeably without necessity
- On line 164, $W_i$ is defined as "utility", but it is then specified as a treatment level $w_i$ on line 170. Also, $y_i(w_i)$ is then specified as observed utility, while $Y_i(W_i)$ is an outcome on line 162.
- The description of Wasserstein distance and projection on lines 209--218 is unclear for readers unfamiliar with the literature.
- As the paper describes how SDP holds in Jiang et al. (2020)'s setting and your setting, on lines 233--248, it is unclear what is the same or different between these settings.
- It seems that you overload notation $p_{\pi_a(X_i}$, $\pi_a(X_i)$, $\pi(X, a)$, and $\mathbb{P}(W_i | X_i, a)$ on lines 166 and 244. $r_i$ on line 235 is also used for this concept.
- There is no Proposition C.1 on line 247, only Lemma C.1. In the proof, on line 930, it is not clear whether C.1 is your own or stated verbatim from Jiang et al. (2020).
- Why is $\mathcal{G}(r, \epsilon)$ written in terms of $\mathcal{X}$ whereas $\mathcal{F}_{r, \epsilon}$ written in terms of $\mathcal{Z}$?
- infimum on line 290 appears to be repeated without significant change
- The discussion about non-convex optimizations algorithms for solving Eq (11) could be made clearer.
- $N \mathcal{R}_{r, \epsilon}(\hat{\mathbb{P}}_N)$ is undefined on line 364.
- Why is DP written in terms of $\mathbb{P}(\mathbf{1}\{\mathbb{P}(W_i = 1 | X_i, S_i) > \tau\} = 1|S_i=1)$, rather than $\mathbb{P}(W_i = 1 | X_i, 1) > \tau)$ on line 238? Can you please clarify this coherence?
- $M(x)$ should be defined outside Theorem 3.1 since it is used throughout the rest of the paper.
- What is $\delta_{x_i}$ on line 287?


Minor:
- Typo in Assumption 1: forgot "(i)" for Uncounfoundedness
- Typo Definition 1: "semiconinuous" should be "semi-continuous"?
- Incomplete sentence on line 209
- Definition 1: might be worthwhile to write $(\pi, \pi') \in \mathbb{Q}_1 \times \mathbb{Q}_2$
- $\mathcal{Z}$ on line 258 should be defined on line 160. Stating $\{0,1\}$ on line 258 is confusing.
- Since the test statistics $t_N$ introduced on line 312 is undefined, it may be worthwhile to discuss how it will be contructed via Theorem 3.2.

**Questions:**

- In figures 2 and 3 (line 420), the test statistic and 95% upper bound are written in terms of $r$, $\epsilon$, and $\theta_1$. Where is this presented in Equation (8) and the discussion on lines 374--377 or 392--397?
- How do the variables defined in Theorem 3.2 and lines 340--349 relate to (P) and Theorem 3.1? It seems that there was a disconnect in this discussion.
- To check my understanding, does the test statistic in Figure 1(a) correspond with the probability that the data does correspond with a $r$-efficient and $\epsilon$-fair distribution, as indexed by the regularization strength?
- Can you clarify what the expectation is taken over in the definition of expected utility $m_w(x, a)$ on line 177, and how this connects to the sample-wise loss discussed on line 433?

---

> ### Author Response · Authors · 2025-11-17
> **Response to Weakness Points**
>
> First, we respectfully disagree with the reviewer for all the beginning 3 weakness points:
>
> - In line 280, we write that the null hypothesis (4) is equivalent to (7). We then define $R_{r,\epsilon}(\hat P_N)$ in (P) (lines 290–298) and discuss it in lines 299–311. Lines 311–313 state that $\eta_{1-\alpha}$ is the quantile of the test statistic, whose construction becomes clear only after deriving its limiting distribution with respect to $R_{r,\epsilon}(\hat P_N)$. The whole subsection 3.1 (strong duality) is for deriving the limiting distribution of the test statistic. Theorem 3.2 provides the stochastic upper bound for the limiting distribution, and lines 374–377 clarify that $N R_{r,\epsilon}(\hat P_N)$ is the test statistic—consistent with lines 311–313 and justifying the definition of $R_{r,\epsilon}(\hat P_N)$ for hypothesis (4)/(7). Thus the exposition is logically structured and clear.
>
> - We indeed use real data: the empirical studies and Figure 1 in Section 4 (lines 428–472) are all based on real data.
>
> - We do provide guidance on generalization beyond the binary case: in the Intro (lines 74–77), we refer readers to Appendix D.2, which details extensions to multiple sensitive groups, and to continuous and multi-level treatments.
>
> ---
> Second, the specific points are mostly misunderstandings with a few minor typos that can be easily corrected:
>
> - We will define infinity sign.
>
> - We will stick to $\nabla$, though in the notations on line 155 we define both $D$ and $\nabla$.
>
> - $Y_i(W_i)$ refers to random variable, $y_i(w_i)$ refers to observed outcome $y_i$ under observed treatment $w_i$ for unit $i$.
>
> - The Wasserstein distance is a well-established concept. Our description closely follows prior published work on fairness hypothesis testing using Wasserstein projection (see the two citations in line 211).
>
> - Lines 228–254 provide a detailed explanation of the foundation and definition of $\epsilon$-SDP. While Jiang et al. apply SDP to general binary classifiers, we use $\epsilon$-SDP for the treatment rule (which can be viewed as a binary classifier deciding which treatment to assign). Could you please clarify which part of our definition is unclear?
>
> - We disagree that these notations can be combined or overloaded. $\pi(X,A)$ follows the notational convention in causal inference for defining propensity score, based upon which $\pi_a(X)$ is a short-hand notation distinguishing the role of $a$ and $x$, $p_{\pi_a(X)}$ refers to the probability density function (line 243), where $X$ is random variable, which is important for quantifying the distances of probability measures. $r_i$ is only used in line 235 once following the introduction of SDP in Jiang et. al, but we are happy to change it to $\pi(X_i,S_i)$, which is easy to be fixed.
>
> - We will change Proposition C.1 to Lemma C.1. Our exposition is clear since we label Lemma C.1 as Prop 1 of Jiang et. al 2020. So the proof follows directly from Jiang et. al, as we specified in line 930.
>
> - It's a typo easy to be fixed: in (3) should be on space $\mathcal Z$ and in (5) should be on space $\mathcal X$.
>
> - In line 290 the first infimum is taken over feasible set (5), the second infimum is taken over the the whole probability space.
>
> - Our contribution is to propose a theoretical hypothesis testing framework. Handling non-convex optimization is inherently complex, and since our focus is theoretical rather than computational, a detailed discussion of this aspect is beyond the scope of the paper.
>
> - $\mathcal R_{r,\epsilon}(\hat P_N)$ is defined in line 290 - 298, N is sample size. So it's already defined.
>
> - In line 238, we follow the presentation style of Jiang et al. when introducing DP and SDP, to help readers easily compare our notation and formulation with theirs.
>
> - We can define $M(x)$ outside Theorem 3.1, but this won't hinder the understanding of the concepts.
>
> - It's standard delta measure that puts all probability mass on $x_i$, will define this in notations.
>
> ---
> Minor:
>
> - Will correct this typo.
>
> - Will correct this typo.
>
> - Sentence is complete, just need to delete redundant word ``and".
>
> - This is fundamental conceptual misunderstanding: $\Gamma(Q_1,Q_2)$ is the set of all joint distributions (couplings) on the product space whose marginals are $Q_1$ and $Q_2$, as we clearly defined in Definition 1 in line 207-208. The notation suggested by reviewer $Q_1\times Q_2$ is a specific element of $\Gamma(Q_1,Q_2)$, which is the independent coupling.
>
> - We could define $\mathcal{Z}$ earlier (e.g., in line 160), but introducing it closer to where it is used helps readers recall the notation more easily.
>
> - We describe the construction of $t_N$ immediately after Theorem 3.2. Its mention in line 312 serves to motivate the subsequent sections: the strong duality results in Sections 3.1 and 3.2 both lead to defining the test statistic. The statistic becomes clear only after deriving its limiting distribution, which we explain right afterward.

---

> ### Author Response · Authors · 2025-11-18
> **Response to questions**
>
> We thank the reviewer for the comments and questions. Due to the character limit, we addressed each specific weakness point in our previous response. Here, we focus on addressing the reviewer’s questions:
>
> - We thank the reviewer for the first question. As stated in Theorem 3.2, we derive a stochastic upper bound and compute the $\eta_{0.95}$ quantile of the right-hand side of Eq. (8) to obtain a conservative test. The right-hand side depends on $r$, $\epsilon$, and $\theta_1$. Incorporating the parametric forms of $\pi_1$, $\pi_0$, and $M(\cdot)$ into this expression is straightforward, and the corresponding plots are provided in Appendix B. We did not include the explicit formula for the test statistic in these parametric cases, since it follows directly from substituting these expressions and adds little conceptual value. However, we are happy to include the full derivation in Appendix B for completeness if the reviewers find it helpful, which again is minor and straightforward.
>
> - We respectfully disagree with the reviewer’s comment regarding a disconnection in the discussion. The notations introduced in lines 340–349 are necessary, as they are integral to Assumption 4 and Theorem 3.2. Moreover, in Theorem 3.2, the term $\mathcal R_{r,\epsilon}(\hat P_N)$ on the left-hand side of Eq. (8) directly relates to Theorem 3.1 — it is exactly equal to the value of (P) by the strong duality result established in Theorem 3.1. Given that the reviewer previously mentioned that the term $N\mathcal R_{r,\epsilon}(\hat P_N)$ was undefined, this may stem from a misunderstanding: the quantity is defined and connected through the theoretical results provided by Theorems 3.1 and 3.2.
>
> - The test statistic indeed is related to but is not exactly equal to the probability that the data does correspond with $r$-efficient and $\epsilon$-fair distribution. Remember that the test statistic is $N \mathcal{R}_{r,\epsilon}(\hat{P}_N)$, which is equal to the sample size multiplied by the objective in (P).
>
> - The conditional expectation is taken with respect to the distribution of potential outcome $Y_i(w)|X_i=x, S_i=a$ (conditioning on $X_i=x, S_i=a$), this is standard in causal inference setting. The loss of unit $i$ is equal to $\ell(W_i,X_i,S_i)$, given treatment $W_i$, covariate $X_i$, sensitive attribute $S_i$ of unit $i$. Since we want to minimize loss, the utility is defined as $-\ell(W_i,X_i,S_i)$, so equivalently we want to maximize the utility. Conditioning on $X_i=x,S_i=a$, the expected utility is defined as $m_w(x,a)=\mathbb E[-\ell(W_i,X_i,S_i)|X_i=x,S_i=a]$.
>
> Please let us know if you have additional questions. Thank you!

---

> ### Author Response · Authors · 2025-12-03
> **All notational issues and typos have been corrected and all marked in blue in the revised version**
>
> We thank the reviewer for pointing out the typos and minor notational issues. We have corrected all of them, and the corresponding changes are highlighted in blue in the revised version of the paper:
>
> - We've defined infinity sign $\infty$ in notation paragraph
> - We've defined delta measure $\delta_x$ in notation paragraph
> - We've sticked to only $D$ for gradient notation
> - We've sticked to $\pi_a(x)$ as the notation for propensity score
> - We've changed "Proposition" to "Lemma"
> - We've corrected the typo: changed $\mathcal Z$ to $\mathcal X$
> - We've corrected the misspelling: "semi-continuous"
> - We've deleted the redundant "and" (now in line 211-212) in the revised version
>
> Additionally, we have clarified all misunderstandings raised by the reviewer in our previous responses. We appreciate the reviewer's comment again, and we hope these clarifications and revisions effectively address all of your concerns.

---

### Official Review · Reviewer_NFVQ · 2025-10-28

**Soundness:** 3
**Presentation:** 2
**Contribution:** 2
**Rating:** 4
**Confidence:** 4

**Summary:**

The paper proposes a framework for assessing fairness-utility trade-offs, for a specific utility function. Specifically, for a threshold on the average reward ($r$) and a violation level ($\epsilon$) of strong demographic parity (the used fairness criterion), the goal is produce a hypothesis test which assesses whether the underlying data distribution comes from family which has an average reward $\geq r$ and a fairness violation $\leq \epsilon$. For the hypothesis testing, an optimal transport formulation is used, and the dual problem of the optimization problem is constructed in order to make the formulation tractable.

**Strengths:**

(S1) The paper deals with an interesting topic of fairness-utility trade-offs, a topic which still does not

(S2) The proposed formulation seems interesting and novel.

(S3) There are some good ingredients in the part related to duality and deriving a test.

**Weaknesses:**

(W1) A core question is what kind of utility functions can be analyzed using the described approach. Here, specifically, two types of utilities are worth distinguishing:

(a) pre-specified, known utility functions where $m_0, m_1$ are given,

(b) unknown utilities, which need to be inferred from data.

This key discussion seems to be largely absent in the manuscript. Specifically, in Line 388, it is stated that “we may require that M(\cdot) be concave”. Even in the case (a), is this a reasonable assumption? For commonly used utilities, such as accuracy, does this hold true? This seems quite important for the applicability of the proposed approach.

For case (b), things seem even more subtle. In the current formulation, is $M()$ assumed to be known perfectly? For instance, in the setting of patient treatment allocation (mentioned in the text), the function $M$ is almost never known, but estimated from data.
Can such cases be handled, where there is inherent uncertainty over $M$?
For instance, is the derivative of $M$ needed for computing $\xi$ values? How can one obtain such a derivative of $M$ in this case?
Also, in these cases, it is unclear whether any convexity/concavity can be expected. It is really important to delineate the scope for which the described hypothesis tests can be described -- currently, this is not done sufficiently well.

(W2) Relating to the previous point, it is unclear whether the potential outcomes framework is needed, especially if only utilities of type (a) can be handled. If so, one could simply say that the utility $Y$ is computed through a known function $f_y (x, a, w)$. The causal semantics may be superfluous if this is the case; potential outcomes seem to be relevant for cases of type (b).

(W3) Another question that comes to mind is computational complexity. It seems that an optimization problem needs to be solved for each $X_i$ — does this become very hard / intractable when there are many distinct values of $X_i$?

(W4) The paper would benefit from better exposition in the start. Adopting a graphical representation (even though not necessarily a causal graph), would be helpful. Covariates $(X, S)$ influence a decision $W$, and $(X, S, W)$ influence the utility $Y$. A graph / visual with $(X, S) \rightarrow W$, $(X, S) \rightarrow Y$, $W \rightarrow Y$ would help ground the mental model.

Part of the issue is that the notation is not quite consistent with other parts of the literature, and this may be worth clarifying; the fairness criterion is placed on $W$, while, usually in the literature, the outcome on which fairness constraints are considered is labeled $Y$. In this work, the utility function is labeled $Y$. This may confuse the reader somewhat.

(W5) The citations for statistical/demographic parity seem incorrect (Dwork et. al. 2012, Agarwal et. al. 2019 are not the correct citations).

(W6) Some of the notation is confusing. In Line 72, $\pi_{S_i}(X_i)$ is defined; in Line 166, this is redefined via $\pi(x, a)$, and then a shorthand is adopted $\pi_a(x)$. This complicates things unnecessarily.

(W7) Second paragraph of Section 1.1 has some overlap with the later parts of the text. The writing could be more tight in this respect.

(W8) The setting description in Section 1.1 would substantially benefit from an example which grounds all of the notions.

(W9) Verifying the ignorability assumption: this part seems unusual; if I understand the appendix correctly, the ignorability criterion is implied by the simple fact that the decision $W$ is based on the observed covariates, since it is a machine learning predictor? If so, saying that this assumption is “tested” seems unusual.


(W10) Assumption 3 — is it reasonable to assume $\pi_1(x) = \pi_0(x)$ for the $x$ that attains a utility $\geq r$? It would be worth reflecting on this, since the condition is not just about $M(x) \geq r$ as mentioned in the text?

Minor:

(W11) Definition 1, typo in semiconinuous

(W12) in Line 213, $c(\cdot, \cdot)$ has $\infty$ appearing in its definition; this seems quite unusual.

(W13) Eq. (3) should $\tilde Q$ be an element of $\mathcal{P}(\mathcal{Z})$ instead of $\mathcal{P}(\mathcal{X})$?

**Questions:**

(Q) Are the choices of $r, \epsilon$ considered to be supplied by the user? Is it easy for the user to decide what level of fairness violation they are willing to tolerate?

(Q) A question about a naive baseline comes to mind: how about quantifying the uncertainty over the expected reward E[Y(W)] and the fairness violation, using bootstrap; and then performing a hypothesis test based on how likely it is that $E^b[Y(W)] \geq r, |SDP^b| \leq \epsilon$ over different bootstrap samples $b$? Is there way of justifying the proposed method over this approach?

---

> ### Author Response · Authors · 2025-11-16
>
> Thank you for your comments!
>
> (W1-a) Our theory doesn't require $M(\cdot)$ to be concave. As stated in the paper, the phrase “we may require that $M(\cdot)$ be concave” appears only in the computation subsection to highlight that concavity enables more efficient computation. When $M(\cdot)$ is non-concave, alternative methods—referenced in lines 389–391—can be used from the non-convex optimization literature.
>
> (W1-b) When utilities are unknown, we assume a known parametric form with unknown parameters. These are estimated from data, allowing derivatives to be computed. Our empirical analysis follows exactly this procedure.
>
> (W2) First, our framework does not require full knowledge of the utility function. Second, we use the potential outcomes framework as a clear and intuitive starting point for defining utilities as functions of $(x,a,w)$. While a fully general formulation is possible, it would obscure the motivation and reduce clarity. As the reviewer notes, our approach readily extends to more general utility specifications, underscoring its flexibility and generality.
>
> (W3) When a parametric form of $M(\cdot)$ is assumed, as we responded to (W1-b), computation becomes easier, as one typically does not need to solve for each $X_i$ individually.
>
> (W4) In the potential outcomes framework, the outcome (utility) is typically denoted by $Y$ and the treatment by $W$. Nonetheless, we are happy to include a graph to improve clarity for readers more familiar with the fairness literature.
>
> (W5) We respectfully disagree, since Dwork et. al. 2012 and Agarwal et. al. 2019 both study statistical parity. Statistical parity is also known as demographic parity (DP) (e.g. see the explanation in Introduction of [1] for reference)
>
> [1] Rychener, Yves, Bahar Taskesen, and Daniel Kuhn. "Metrizing fairness." arXiv preprint arXiv:2205.15049 (2022).
>
> (W6) In line 72, we introduce $\pi_{S_i}(X_i)$ to summarize the main results in advance. The expression $\pi(x,a)$ appears only once (line 166) to follow the potential outcomes convention of writing the propensity score as a function of both $x,a$. We then state (line 168) that $\pi_a(x):=\pi(x,a)$ and use this notation consistently. We can define $\pi(x,a)=\pi_a(x)$ directly if preferred, though this may confuse readers familiar with causal inference conventions.
>
> (W7) Section 1.1 serves as an overview of the main results. While some notions and concepts are introduced informally for readability, we later define all notations and concepts formally to ensure rigor and coherence.
>
> (W8) Although examples of the hypothesis testing framework are given in lines 299–310, we will add another in Section 1.1 as suggested. Consider consumer lending, where $Y_i(W_i)$ is the company’s profit under decision $W_i \in {0,1}$ (lend or not). Here, $S_i$ is the borrower’s gender or race, $X_i$ their demographics, and $\pi(X_i,S_i)$ the loan approval probability.
>
> (W9) We did not claim to have “tested” the assumption; instead we say that it holds by design in our empirical setting, as the study design satisfies the unconfoundedness condition.
>
> (W10) We believe Assumption 3 is reasonable. If it failed, achieving the desired utility would require treating everyone unfairly—an overly restrictive case implying a need to adjust $r$, since it could potentially be socially harmful.
>
> (W11), (W13) We will correct the typos.
>
> (W12) These definitions are exactly the same as Taskesen et al. 2021, Si et. al 2021 as we specified in line 211, implying that it is not unusual.
>
> (Q1) The choice of $\epsilon$ and $r$ are chosen by the user. As discussed in lines 303–309, the ease of selecting these parameters depends on the specific application context. In the example provided in the paper, this choice is straightforward: when the utility represents the revenue of a consumer lender, $r$ corresponds to the target expected revenue the lender aims to achieve, while $\epsilon$ quantifies the acceptable level of disparity in approval rates across sensitive groups (e.g., male vs. female). For instance, one might set $\epsilon = 0.05$ based on an industry standard or organizational fairness policy.
>
> (Q2) Great question! Our test statistic is a Wasserstein projection distance that jointly enforces both the utility and relaxed SDP constraints. This projection formulation captures the interaction between the two conditions and the non-smooth boundary of the feasible set. In contrast, a simple bootstrap treats the two quantities as independent scalars and ignores the geometry induced by the projection. Because the projection involves minimization and indicator operations, it is a non-smooth, non-parametric functional, for which naive bootstrap methods are often invalid. By deriving the statistic’s non-standard limiting law (Theorem 3.2) and using the Gaussian-based resampling scheme in Sec. 3.3, our method properly calibrates the test and guarantees asymptotic Type-I error control even in this non-regular setup.

---

> > ### Comment · Reviewer_NFVQ · 2025-11-22
> >
> > I thank the authors for the detailed response. However, some important concerns remain.
> >
> > (W1a) I understand that the theory does not require $M$ to be concave. My question was about whether non-convex functions actually work in practice. While it is convenient to defer the issue to non-convex optimization literature, it does not really answer the question of whether the approach is practicable (computationally) for interesting non-convex functions. For instance, the experimental evaluation does not look into this at all; and the discussion of which non-convex functions would be interesting is missing.
> >
> > (W1b) Parametric forms for the utility are a strong assumption. Furthermore, the whole framing of which utilities can be used is quite opaque. These things should be crystal-clear after reading the paper.
> >
> > Furthermore, the empirical analysis is very limited, and referring to it is not at all helpful. In simulated data, you are using a known linear utility. For real-data experiments, you just write $m_w(x,a)$, and point Appendix E, which has no information whatsoever on what this utility is.
> >
> > A fundamental concern in my question is also whether _stochastic_ utilities can be handled. This is essential in many real-world applications, where a decision influences a random outcome.
> >
> > (W2)
> >
> > > We use the potential outcomes framework as a clear and intuitive starting point for defining utilities as functions of $(x, a, w$).
> >
> > As already mentioned, if your utility function is not random, it is unclear whether the PO framework is even needed (since in this case the potential outcome can be defined via $m$ directly and the ignorability assumption seems to be trivially satisfied anyway).
> >
> > > As the reviewer notes, our approach readily extends to more general utility specifications, underscoring its flexibility and generality.
> >
> > I am making no such notes or statements.
> >
> > (W5) Thanks, I am not disputing that these works are using DP / SP. I am pointing to the fact that these notions have been used in the same context in older literature too, e.g., [1], which is sometimes used as a reference.
> >
> > [1] Darlington, Richard B. "Another look at “cultural fairness” 1." Journal of educational measurement 8.2 (1971): 71-82.
> >
> > (W6)
> >
> > > The expression $\pi(x,a)$ appears only once (line 166)
> >
> > This is exactly the issue I was raising.
> >
> > (W9)
> >
> > > We did not claim to have “tested” the assumption.
> >
> > Thanks. You have used the word _verify_, which sounds as if some kind of testing is performed. I am unsure if a separate appendix for saying the assumption holds by construction is needed.
> >
> > (W12) While I appreciate the same definition may have appeared elsewhere, this does not imply that it is good or precise notation.
> >
> > (Q2) Are you making a formal statement? If you were able to either (1) make such a formal statement; (2) show empirically how your approach compares to bootstrap, and why it is better in a real-world, this would strengthen the paper significantly. Current empirical evaluation does not even take bootstrap into consideration, which is another limitation.
> >
> > To conclude, as mentioned already, the paper has some good ingredients. However, it also has quite significant space for improvement. I maintain my score.

---

> ### Author Response · Authors · 2025-11-27
> **Additional numerical studies and responses to reviewer's remaining concerns**
>
> We thank the reviewer for the reply.
>
> ---
>
> **(Q2) We have conducted additional experiments comparing our method vs. bootstrap:**
>
> - We find that bootstrap is consistently more conservative than our method for all values of $(\epsilon,r)$: Whenever our test rejects, the bootstrap test also rejects; moreover, the bootstrap test rejects in additional cases where our method does not. Recall that Theorem 3.2 establishes that our test is conservative, as it is constructed using a stochastic upper bound. The naive bootstrap is even more conservative. These observations further underscore the validity and usefulness of our proposed method.
>
> - We have added additional numerical results in Section 4 and Appendix F, together with formal explanations for why this phenomenon arises in lines 453-467. (marked in blue in the most up-to-date version)
>
> ---
>
> (W1a):
>
> - Regarding the reviewer’s concern about the non-concavity of $M$: While this issue stems from the inherent non-convexity of the optimization problem, it is a challenging and broad topic on its own. It lies outside the central focus of a theoretical paper on hypothesis testing under a fairness–utility trade-off. We include a preliminary discussion of computational aspects in Section 3.3 and respectfully believe that this treatment is appropriate for the paper’s scope and contributions.
>
> - **Further, we want to emphasize that our paper is, to the best of our knowledge, the first to implement a hypothesis test of fairness under a fairness–utility trade-off framework via Wasserstein projection method**. The only existing works on hypothesis testing of fairness via Wasserstein projection are Taskesen et al. (2021), Si et al. (2021) — both of which we cite in line 211 — and one unpublished arXiv preprint [1].
>
> [1] Li, W., Park, Y. P., \& Duc, K. D. (2025). Wasserstein projection distance for fairness testing of regression models. arXiv preprint arXiv:2510.04114.
>
> ---
>
> (W1b): We respectfully disagree with the reviewer on this point.
>
> - First, Parametric forms of utility function is common in Econ and CS literature even in recent years, to name a few [1], [2], [3], etc.
>
> [1] Yao, Fan, et al. "Unveiling user satisfaction and creator productivity trade-offs in recommendation platforms." Advances in Neural Information Processing Systems 37 (2024): 86958-86984.
>
> [2] Chen, Yiling, et al. "Bias detection via signaling." Advances in Neural Information Processing Systems 37 (2024): 69120-69143.
>
> [3] Yao, Fan, et al. "How Bad is Top-$ K $ Recommendation under Competing Content Creators?." International Conference on Machine Learning. PMLR, 2023.
>
> - Second, our real-data experiments follow the same empirical settings used in the most closely related papers (Taskesen et al., 2021; Si et al., 2021). Moreover, the utility function cannot be inferred from the dataset alone; it must be defined by the decision maker based on the specific empirical context. Additionally, our paper is primarily a theoretical contribution --- the empirical and numerical studies are included to verify the theoretical results, which they successfully do. For these reasons, we respectfully disagree with the reviewer’s assertion that our empirical study is not helpful.
>
> For the above reasons, we respectfully hold that the concerns raised by the reviewer are not central to the main goals of this paper and should not diminish the conceptual and methodological contributions of our proposed fairness-testing framework.
>
> ---
>
> (W2): As explained in our previous response, we use the potential outcomes framework as a clear and intuitive starting point for defining utilities as functions. While a fully general formulation is certainly possible, it would obscure the core motivation and reduce clarity. The reviewer’s comment implicitly acknowledges that our framework can be generalized further; however, our aim is to present a clean and well-motivated formulation, supported by concrete examples in the paper. We respectfully view this comment as a minor notational preference that is not central to the focus of our work.
>
> ---
>
> (W5): We respectfully disagree with the reviewer on this point. We believe we have properly cited the relevant references, and the reference mentioned in the reviewer’s final comment does not discuss SP/DP either.
>
> ---
>
> The points raised in (W6), (W9), and (W12) concern minor notation or formatting issues. We have addressed all of them in the revised version of our paper. Additionally, we have also added a graphic illustration showing (X,S) impacts W, (X,S,W) impacts Y in lines 173-183, as the reviewer suggests in the earliest comment.  (all marked in blue in the most up-to-date version)

---

### Meta-Review · Area_Chair_AZLi · 2025-12-26

**Summary:**

This paper presents a statistical hypothesis testing framework that jointly evaluates approximate fairness and utility. specifically, they proposed a Wasserstein projection-based test.

the scores for this submission 6444 where the majority of the reviewers did not vote for its acceptance since there were various concerns bout the current version.

**Reviewer Concerns:**

Reviewer NFVQ has major concern for which I agree with them:

the paper have a vague statement  "we may require that M(\cdot) be concave"; Reviewer NFVQ questioned whether non-convex functions actually work in practice. While it is convenient to defer the issue to non-convex optimization literature, it does not really answer the question of whether the approach is practicable (computationally) for interesting non-convex functions. For instance, the experimental evaluation does not look into this at all; and the discussion of which non-convex functions would be interesting is missing. The authors' response is not satisfactory.

1. Reviewer NFVQ : In the current formulation, is
 assumed to be known perfectly? the author's response seems to unsatisfactory according to the further comments from the reviewer.

2. Reviewer uyZP: the paper’s presentation that could be improved, particularly with respect to clarity and reproducibility.

 3. Reviewer Yb7P who gave the highest score 6 raised the concerns in the rebuttal: i) limited empirical support since it is still difficult to evaluate statistical power, robustness, or practical relevance; ii) Practicality and scalability

4.  Reviewer yZVT expressed concerns on its  practical significance and limited insight into how such findings inform model design or real-world decision-making. they mentioned that the work mainly contributes a statistical testing procedure rather than a learning algorithm. This makes me a bit unsure about its fit for ICLR.



Based on the overall recommendation from the reviewers, I CAN NOT recommend its acceptance.

**Reviewer Scores:**

NA

---

### Decision · Program_Chairs · 2026-01-26

Reject